# NRG1/ErbB signalling controls the dialogue between macrophages and neural crest-derived cells during zebrafish fin regeneration

Béryl Laplace-Builhé [1], Audrey Barthelaix[1], Said Assou[1], Candice Bohaud[1], Marine Pratlong[2], Dany Severac[2], Gautier Tejedor [1], Patricia Luz-Crawford[3,4], Mai Nguyen-Chi[5], Marc Mathieu[1], Christian Jorgensen[1,6] & Farida Djouad[1✉]

Fish species, such as zebrafish (*Danio rerio*), can regenerate their appendages after amputation through the formation of a heterogeneous cellular structure named blastema. Here, by combining live imaging of triple transgenic zebrafish embryos and single-cell RNA sequencing we established a detailed cell atlas of the regenerating caudal fin in zebrafish larvae. We confirmed the presence of macrophage subsets that govern zebrafish fin regeneration, and identified a *foxd3*-positive cell population within the regenerating fin. Genetic depletion of these *foxd3*-positive neural crest-derived cells (NCdC) showed that they are involved in blastema formation and caudal fin regeneration. Finally, chemical inhibition and transcriptomic analysis demonstrated that these *foxd3*-positive cells regulate macrophage recruitment and polarization through the NRG1/ErbB pathway. Here, we show the diversity of the cells required for blastema formation, identify a discrete *foxd3*-positive NCdC population, and reveal the critical function of the NRG1/ErbB pathway in controlling the dialogue between macrophages and NCdC.

[1] IRMB, Univ Montpellier, INSERM, Montpellier, France. [2] MGX, BCM, Univ Montpellier, CNRS, INSERM, Montpellier, France. [3] Laboratorio de Inmunología Celular y Molecular, Facultad de Medicina, Universidad de los Andes, Santiago, Chile. [4] IMPACT, Center of Interventional Medicine for Precision and Advanced Cellular Therapy, Santiago, Chile. [5] LPHI, Univ Montpellier, CNRS, Montpellier, France. [6] CHU Montpellier, Montpellier, France. ✉email: farida.djouad@inserm.fr

Unlike adult mammals, urodele amphibians and fish species, such as zebrafish (*Danio rerio*), can regenerate entire parts of their body, including limbs and fins[1,2], through a process referred to as epimorphic regeneration. This process involves the well-orchestrated restoration of multiple tissues and depends on the formation of a structure named blastema. Although for a long time, the blastema was considered a homogeneous mass of multipotent cells, it is now acknowledged that it is composed of heterogeneous, highly proliferative, and dynamic lineage-restricted progenitor cell types[3–7]. The list of cell types and factors involved in blastema formation and appendage regeneration has been expanded over the last years, but comparatively little is known about the cells and genes engaged upon tissue amputation in vertebrates.

To overcome this lack of knowledge, single-cell RNA sequencing (scRNA-seq) and lineage tracing experiments have been recently combined to study the axolotl limb blastema composition[8,9]. These studies allowed clearly demonstrating that the heterogeneous fibroblast population of the blastema loses its adult features, adopts a multipotent skeletal progenitor phenotype, and expresses genes of the developmental-like state[8,9]. Furthermore, they confirmed the presence of muscle satellite cells, fibroblasts, and macrophages in the regenerating axolotl limbs, and also identified different cell types[9]. Altogether these studies revealed that single-cell transcriptome profiling provides a more detailed view of the blastema structure. It is now important to use this technology in the context of epimorphic regeneration in other vertebrates and at different developmental stages in order to find common features and understand how this regenerative outcome is regulated. This is especially of interest for developing innovative approaches for regenerative therapies. To this aim, a critical step forward would be to determine cell type-specific functions in the blastema of regenerating organisms.

Recently, a single-cell transcriptomic analysis performed at different time points during caudal fin regeneration in adult zebrafish revealed the cellular diversity of regenerating tissues and single-cell transcriptomic dynamics[10]. However, neither myeloid cells nor nerve or cells associated with nerves were identified in this study, despite their crucial role in epimorphic regeneration[11–13]. Specifically, in adult axolotls, macrophages are required for limb regeneration[14,15]. Similarly, in adult zebrafish, macrophage genetic depletion during the entire regeneration process impairs blastema cell proliferation and caudal fin regeneration, while their depletion during the tissue outgrowth phase affects the caudal fin morphology[16]. Using zebrafish larvae, we confirmed the presence of macrophages within the regenerating caudal fin fold and we observed an early and transient accumulation of pro-inflammatory macrophages[17]. Early recruited pro-inflammatory macrophages provides the accurate TNFα signal to prime blastema cell proliferation and regeneration in zebrafish[17]. However, the detailed cellular and molecular mechanisms responsible for macrophage recruitment and activation during epimorphic regeneration in zebrafish have not been elucidated yet. Moreover, Schwann cell (SC) precursors are neural crest (NC)-derived cells (NCdC) associated with almost all nerve fibers. These cells release trophic factors that promote blastemal cell proliferation and epimorphic regeneration[18,19]. In zebrafish, NCdC contribution has recently been demonstrated during cardiac development and regeneration[20,21], but has never been investigated in appendage regeneration. Interestingly, after nerve injury, SC release factors that favour pro-inflammatory macrophage recruitment and polarization towards an anti-inflammatory phenotype[22,23]. These studies suggest that SC might play a pivotal role in the tight regulation of the immune response required for regeneration.

Macrophage-mediated immune response and NCdC paracrine actions have been described separately in the context of regeneration. Therefore, there is a crucial need for a model of appendage regeneration in vertebrates that would integrate these two cell types to study their functional interactions.

In this work, scRNA-seq of cells from intact and regenerating caudal fin fold of zebrafish larvae allowed us to comprehensively describe the blastemal cell heterogeneity and generate an atlas of the cell types involved in regeneration. We identified cell populations within the blastema of regenerating zebrafish larvae, and demonstrated the presence of an orchestrator cell population that regulates the macrophage pro-regenerative response through the NRG1/ErbB signalling pathway. Collectively, these data have important implications for understanding epimorphic regeneration in vertebrates and for regenerative medicine.

## Results

### Identification of the different cell populations in the regenerating caudal fin fold by scRNA-seq.
Caudal fin fold amputation in 3 days post fertilization (dpf) larvae induces a robust regeneration of the missing tissues within 3 days through the formation of the blastema at 24 h post-amputation (hpA). To determine the cellular and transcriptomic profiles of intact (uninjured) and regenerating caudal fin fold samples at 24 hpA, we used scRNA-seq and the Chromium system (10X Genomics platform). This system employs the microdroplet technology to isolate individual cells, followed by next-generation sequencing, analysis and visualization of the single-cell datasets with the Cloud software (Fig. 1a). The two-dimensional (2D) distribution profiles obtained by Uniform Manifold Approximation and Projection (UMAP) revealed seven distinct cell clusters (K-mean = 7) in the regenerating fin fold (Fig. 1b). Then, we generated the lists of the top 100 upregulated transcripts (filtered by FDR < 0.05 and ranked by fold change) in each cluster of the regenerating fin sample (Fig. 1b and Supplementary Data 1). Hierarchical clustering based on these differentially expressed genes showed specific signatures in each cluster (Fig. 1c). We used well-structured lists of markers to assign a cell identity to each cluster. We assessed the robustness of these markers by reviewing the literature (Supplementary Data 2). In the regenerating fin sample, we focused on cluster 7 because it included cells that expressed neuron and glia markers, such as *CD59*, *tfap2* and *egr2* (Supplementary Data 1). In cells of this cluster, *hmx3a* (a gene encoding a protein predicted to have sequence-specific DNA binding activity) and *irg1l* (a gene encoding a protein involved in the inflammatory response to wounding) also were upregulated, suggesting that this cell population might specifically respond to amputation (Fig. 1d-f). Moreover, in cluster 7, we identified cells that overexpressed *foxd3*, *sox10,* and *erg2b* (Fig. 1f), and the presence of a specific signature in *foxd3*+, *sox10*+, and *erg2b*+ cells (Fig. 1g). We then confirmed, in cluster 6, the presence of myeloid cells that express *card9*, *il1b*, and *spia*. Cluster 5 contained cells that strongly expressed *nr2f1a* (involved in circulatory system development and endothelial cell proliferation) and genes involved in cell cycle and proliferation (*aurkb*, *top2a*, *mki67*). Also, we confirmed *fgf8a* expression in apical epithelium cap cells (cluster 4), and highlighted the expression of matrix metalloproteinase-encoding genes (*mmp9* and *mmp13a*). Cluster 3 corresponded to mesenchymal cells and epidermis/mesenchymal cells and expressed mesenchymal cell markers (*pdgfrl*, *twist1b*, *twist3*). Clusters 2 and 3 were also enriched in extracellular matrix protein-encoding genes (*col5a2a*, *col1a2a*). In cluster 1 that included epidermis cells, *cldne*, *evpla* and *agr1* were strongly expressed. However, we could not formally identify cells in cluster 2 because they did not specifically express any typical cell/tissue marker. We hypothesized that this cluster comprised epidermis cells because they expressed *lgals1l1* or *krt91*.

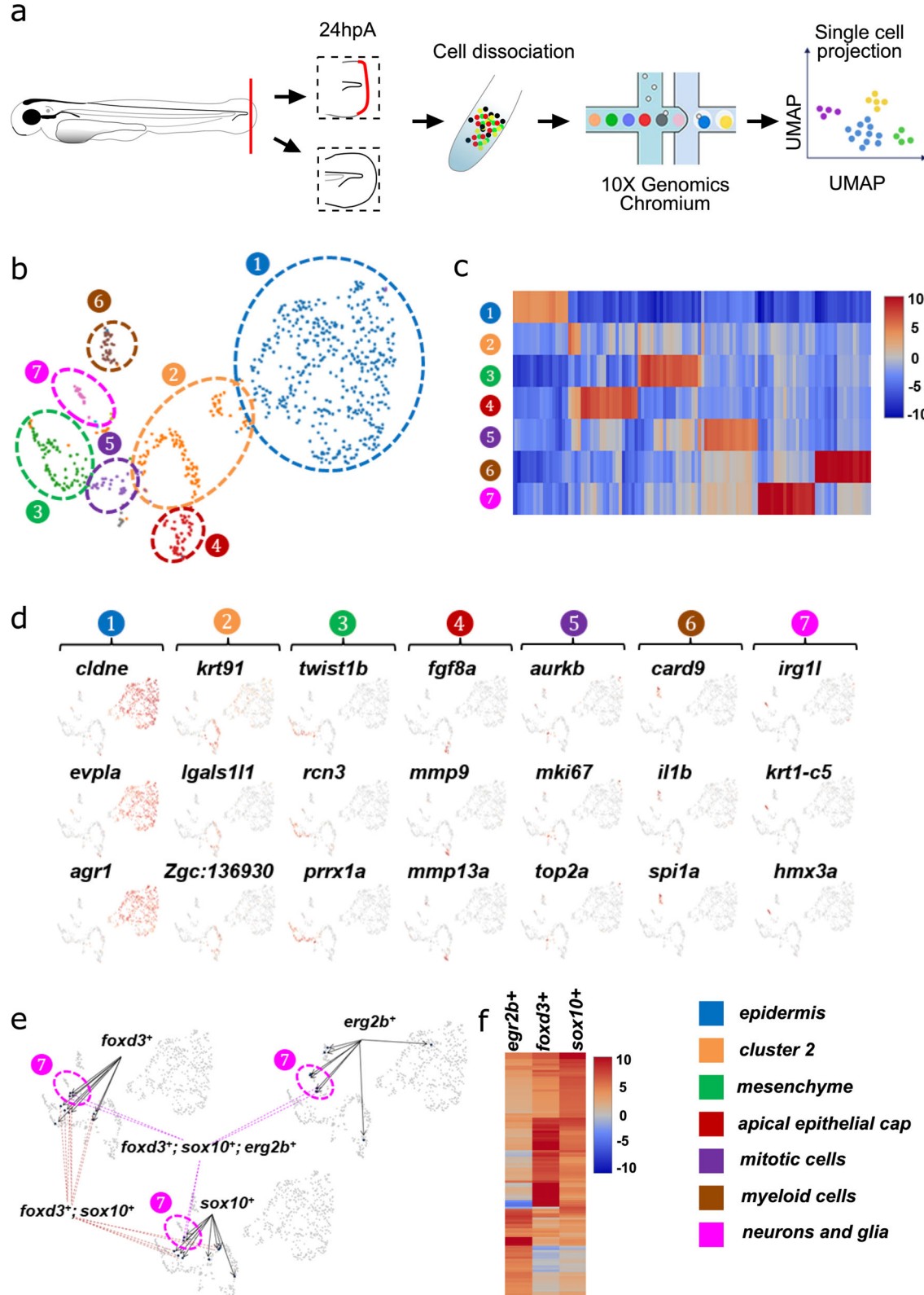

**Fig. 1 Characterization of cells in the zebrafish blastema by scRNA-seq. a** scRNA-seq experiment design. **b** Uniform Manifold Approximation and Projection (UMAP) plots visualizing RNA-sequencing data of single cells from the cut caudal fin fold. Seven different cell clusters were identified **c** Heatmap displaying the genes that are differentially expressed in the seven clusters. Red, upregulated genes, and blue, downregulated genes. **d** UMAP representation of the expression (low to high, grey to red) of cell markers in the different clusters. **e** *Foxd3, sox10,* and *egr2b* expression in cluster 7 (low to high expression, grey to blue). **f** Heatmap of the gene expression profile of the *foxd3+, sox10+,* and *egr2b+* cell populations.

Moreover, we used the Cell Ranger software to aggregate and compare the expression data in the amputated fin and in the intact control (Supplementary Fig. 1a). The comprehensive list of the 100 differentially expressed genes is presented in Supplementary Data 3. Then, we used marker genes (Supplementary Data 2) and the UMAP method to distinguish cell subpopulations in the uninjured and injured samples. This analysis showed that cell clusters were quite comparable between conditions. This is in line with a similar study carried out in *Xenopus laevis*[24] showing that wound epidermis is not a novel cell state, but a redeployment of the embryonic apical epidermal ridge. As our study relied on developing zebrafish larvae and on the basis of the similitudes between clusters of the two studies, we propose the same hypothesis. The most striking difference was the strong increase of the myeloid cell cluster in the injured condition compared with uninjured samples. This result was expected because myeloid cells are recruited after amputation (Supplementary Fig. 1c). Finally, analysis of the uninjured sample datasets (Supplementary Fig. 1d) revealed the presence of another glia/neuron subpopulation type in cluster 6 and some muscle markers in cluster 4, indicating light contamination from the distal trunk muscle cells in this sample (Supplementary Fig. 1e-f). The lists of the top 100 upregulated transcripts in each cluster of the intact fin sample are detailed in Supplementary Data 4. Altogether, these results demonstrated the cell type heterogeneity of the regenerating caudal fin fold of zebrafish larvae and identified a cell population that corresponds to *foxd3*+*sox10*+*erg2b*+NCdC.

**In the regenerating caudal fin fold, *foxd3*+NCdC exhibit morphological and phenotypic changes**. The scRNA-seq analysis revealed the existence of cells that express NC cell markers (cluster 7), including *foxd3*. NC cells express several transcription factors, such as *foxd3* and *sox10*, that display pivotal functions during NC development[24]. *Foxd3* is expressed in pre-migratory NC cells, and also in some migrating and differentiating NC cells[25]. In urodele amphibians, NCdC release trophic factors that promote blastemal cell proliferation and epimorphic regeneration[18]. In zebrafish, NCdC contribute to cardiac development and regeneration[20,21], but their role in appendage regeneration is not known. To determine whether the *foxd3*+ cell population in the blastema was involved in caudal fin fold regeneration in zebrafish larvae, we first analysed *foxd3*+ NCdC presence and behaviour by confocal microscopy. First, analysis of *Tg(foxd3:eGFP-F)* zebrafish larvae showed the presence of eGFP+ cells in the developing fin bud at the 11- and 14-somite stages (Fig. 2a) and in the caudal fin fold at day 3 post-fertilization (Fig. 2b, c). Second, to determine the relative proportion of *foxd3*+ cells in the intact fin fold mesenchyme at 3 dpf and during regeneration (Fig. 2d), we used the triple transgenic zebrafish larvae *Tg(foxd3:eGFP-F/rcn3:Gal4/UAS:mCherry)* to track NCdC (green) and mesenchymal cells (red)[12,26]. Confocal microscopy imaging showed the presence of *foxd3*+ NCdC in the caudal fin fold mesenchyme at 3 dpf (Fig. 2e; Supplementary Fig. 2a) and the increased number of *foxd3*+*rcn3*+mesenchymal cells (yellow cells) at 6 hpA and 24 hpA in the caudal fin fold (Fig. 2e, Supplementary Fig. 2c, d). This observation could indicate the presence of an NCdC population that expresses mesenchymal cell markers within the regenerating fin, suggesting NCdC reprogramming or differentiation into mesenchymal cells during regeneration. FACS analysis of the regenerating caudal fin fold at 6 and 24 hpA confirmed this hypothesis (Fig. 2f-i). The percentage of *foxd3*+NCdC was not significantly different in the intact and regenerating caudal fin fold at 6 and 24 hpA (Fig. 2h). Conversely, the percentage of *mCherry*+ mesenchymal cells (Fig. 2g) and of *mCherry*+*eGFP*+ mesenchymal cells (Fig. 2i) was

significantly higher in the regenerating caudal fin fold at 24 hpA. Finally, at 6 hpA, roundness (Supplementary Fig. 2e) and circularity (Supplementary Fig. 2f) of *foxd3*+*rcn3*+ cells were increased and the cell elongation factor (Supplementary Fig. 2g) was decreased at the wound site compared with control caudal fin fold (uncut). Altogether, our results suggest that these morphological changes accompany the phenotypic and functional modifications of *foxd3*+ cells during regeneration.

**In the blastema, mesenchymal cells proliferate in contact with *foxd3*+NCdC**. To better understand the increase of *foxd3*+*rcn3*+ and *rcn3*+ cells upon caudal fin fold amputation, we assessed the proliferation rate of mesenchymal and non-mesenchymal cell populations throughout regeneration in *Tg(rcn3:Gal4/UAS:mCherry)* and *Tg(foxd3:eGFP-F)* larvae by immunodetection of phosphorylated histone 3 (PH3) that labels proliferative cells. At the wound site, the proliferation rate of PH3+*rcn3*+ mesenchymal cells was significantly higher than that of PH3+*rcn3*− cells at 6 and 24 hpA (Fig. 3a). Conversely, upon amputation, *foxd3*+ cells did not proliferate (Fig. 3a and Supplementary Fig. 3a). Moreover, all proliferating *rcn3*+ cells were physically close to *foxd3*+NCdC, as revealed by 4D confocal microscopy at 6 hpA in *Tg(foxd3:eGFP-F/rcn3:Gal4/UAS:mCherry)* larvae (Fig. 3b and Movie 1). This observation was confirmed by counting PH3+ cells in the contact area with *foxd3*+ cells (Supplementary Fig. 3b-c). Indeed, the percentage of PH3+ cells in contact with *foxd3*+ cells was higher than that of PH3+ cells alone in the cut condition (Supplementary Fig. 3b), but not in the uncut condition (Supplementary Fig. 3c). These results confirmed the presence of *foxd3*+NCdC in the regenerating caudal fin fold, and identified phenotypic and morphological changes in caudal fin fold mesenchymal cells during regeneration. Moreover, some of this newly identified *foxd3*+NCdC interact tightly with proliferating mesenchymal cells.

***Foxd3*+NCdC are required for zebrafish caudal fin fold regeneration**. The observed morphologic and phenotypic changes in *foxd3*+NCdC and their physical interactions with proliferative mesenchymal cells during blastema formation suggested that *foxd3*+ NCdC could play a role in blastema cell proliferation and formation during caudal fin fold regeneration. To address this hypothesis, we partially depleted *foxd3*+ NCdC using morpholino-mediated knockdown of *foxd3*[27]. Foxd3 governs the expression of other critical transcription factors, such as *snail* and *sox10*, in nascent NC cells and promotes NC cell subset survival[28]. Moreover, the zebrafish *sym 1* mutant, which possesses a functional null allele of the zebrafish *foxd3* gene, exhibits a normal number of pre-migratory NC cells, but a reduced number of cells that delaminate from the neural tube, and a defect in *snail* and *sox10* expression levels[29]. The disappearance of NC-derivative structures, such jaw and otic vesicle, and the reduced number of pigment cells that appeared immature in *foxd3* morphants (MO*foxd3*) compared with control morphants (MO*ctl*) (Supplementary Fig. 3d) confirmed the loss of Foxd3 activity. Moreover, we used the *Tg(foxd3:mCherry)ct110* heterozygote line that expresses the FoxD3-mCherry fusion protein[30] to validate the efficacy of the *foxd3* morpholino *(MOfoxd3)* (Supplementary Fig. 3e). Confocal microscopy showed that, in 3 dpf embryos injected with MO*foxd3*, the number of *foxd3*+NCdC within the mesenchyme was reduced (Fig. 3c). Caudal fin fold regeneration involves the critical step of blastemal cell proliferation in the area beneath the apical epithelial cap (AEC) from 6 hpA, followed by cell proliferation propagation to more proximal regions from 24 hpA[31]. Analysis of cell proliferation by PH3 immunodetection in the 24 hpA blastema of MO*ctl*- and MO*foxd3*-injected

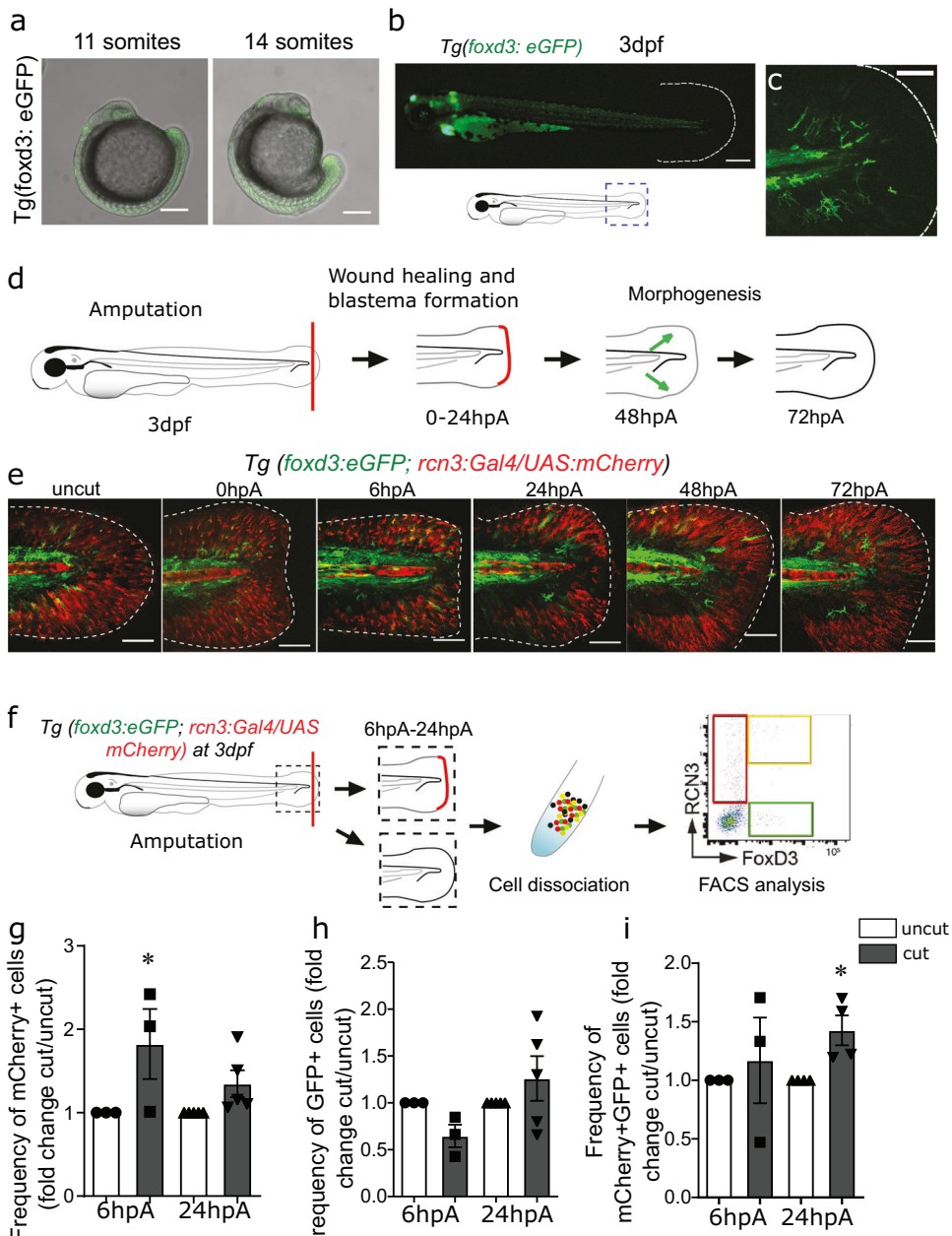

**Fig. 2 Foxd3+NCdC in the developing and regenerating caudal fin fold mesenchyme are located next to proliferative rcn3+ mesenchymal cells.**
**a** Confocal images of *Tg(foxd3:eGFP-F)* zebrafish embryos at 11 somites and 14 somites (Scale bars = 300 μm). **b, c** Representative fluorescent images of *Tg(foxd3:eGFP-F)* 3 dpf larvae (Scale bar = 400 μm. (**b**) scale bar = 80 μm (**c**) **d** Schematic representation of caudal fin fold regeneration after amputation of 3 dpf larvae. **e** Representative confocal microscopy images of *Tg(foxd3:eGFP-F/rcn3:Gal4/UAS:mCherry)* larvae during regeneration (Scale bars = 80 μm, representative from 5 biologically independent larvae examined over 3 independent experiments). **f** Schematic representation of the fluorescent-activated flow cytometry analysis of *eGFP+* and *mCherry+* cells from caudal fin fold of *Tg(foxd3:eGFP-F/rcn3:Gal4/UAS:mCherry)* larvae at 6 and 24 hpA. Data were analysed using FlowJo v10. **g** Frequency of *mCherry+* cell number in the caudal fin fold at 6 and 24 hpA relative to the age-matched uninjured controls. **h** Frequency of *eGFP+* cell number in the caudal fin fold at 6 and 24 hpA relative to the age-matched uninjured controls (fold change). **i** Frequency of *mCherry+eGFP+* cell number in the caudal fin fold at 6 and 24 hpA relative to the age-matched uninjured controls. **g-i** Graphs represent the mean value ± SEM, one-tailed Wilcoxon test was performed, *p < 0.05, p = 0.05 (**g**), p = 0,0143 (**i**), n = 50–300 larvae per group from 5 independent experiments.

zebrafish larvae revealed that cell proliferation rate was significantly lower in regenerating *foxd3* morphants (MO*foxd3*) compared with controls (MO*ctl*) (Fig. 3d) and consequently caudal fin fold growth was significantly reduced, compared with embryos injected with control morpholino (MO*ctl*) (Fig. 3e). This was correlated with the altered expression pattern of *junbl*, a blastemal marker[31], observed by in situ hybridization (Supplementary Fig. 3f). To confirm NCdC role in blastemal cell

proliferation, we used homozygous *Tg(foxd3:mCherry)^{ct110}* 3 dpf larvae that present similar defects in NC cell development as *foxd3* morphants[30]. Compared with control zebrafish larvae (wild type, WT), fin fold outgrowth was severely impaired in *Tg(foxd3:mCherry)^{ct110}* mutant larvae at 72 hpA (Fig. 3h). This impairment of the regenerative capacity was associated with a significant decrease of PH3+ cells in the blastema compared with WT zebrafish (Fig. 3g), as well as a decrease of *junbl* expression at

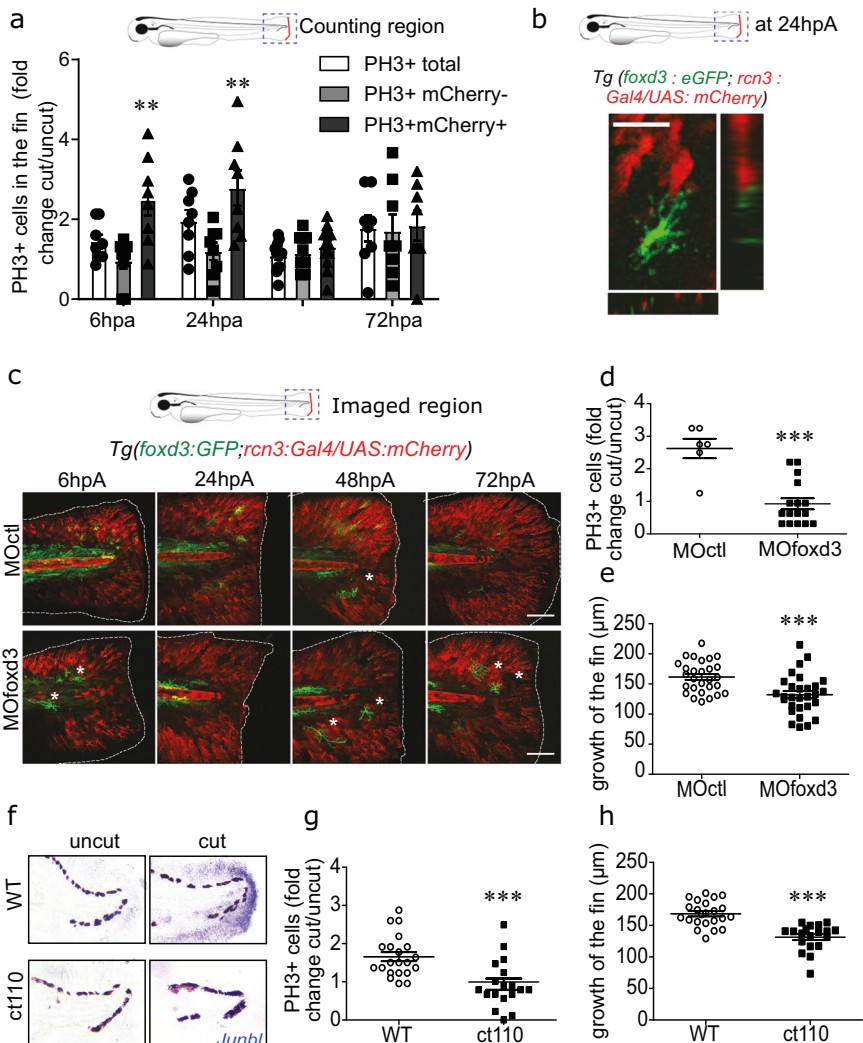

**Fig. 3 _Foxd3_+ NCdC are required for blastemal cell proliferation and zebrafish caudal fin fold regeneration. a** Quantification of cell proliferation in the blastema of _Tg(rcn3:Gal4/UAS:mCherry)_ larvae from 6hpA to 72 hpA. Mitotic cells were detected using an anti-PH3 antibody. Data are shown as fold change relative to the age-matched uninjured controls, and are the mean ± SEM, $n = 8$ (for the 6, 24, 72 hpA time points), $n = 12$ (for the 48 hpA time point), **$p < 0.01$. **b** Image of a time-lapse z-stack sequence of a _Tg(foxd3:eGFP-F/rcn3:Gal4/UAS:mCherry)_ larvae at 24 hpA with x and y projections using the Fiji software (Scale bar = 8 μm). **c** Confocal images of caudal fin fold of _Tg(foxd3:eGFP-F/rcn3:Gal4/UAS:mCherry)_ larvae injected with MO_ctl_ or MO_foxd3_ at 3 dpf, and then from 6 hpA to 72 hpA (asterisks show pigments, scale bars = 80 μm, representative of $n = 5$ larvae from 3 independent experiments). **d** Blastemal cell proliferation in _foxd3_ and control morphants was assessed using an anti-PH3 antibody at 24 hpA. Data are shown as fold change relative to the age-matched uninjured controls, and error bars are the SEM, $n = 6$ (for the control group) and $n = 16$ (for the _foxd3_ morphants) biologically independent larvae, one-tailed Mann-Whitney test was performed, $p = 0.0007$, ***$p < 0.001$. **e** Quantification of caudal fin fold length in _foxd3_ and control morphants at 72hpA. Error bars are the SEM, $n = 21$ biologically independent larvae, one-tailed Mann-Whitney test was performed, $p = 0.0002$, ***$p < 0.001$. **f** The mRNA expression of the blastemal marker _junb-l_ (blue) was detected by in situ hybridization at 24 hpA in control and amputated fin folds from 4 dpf wild type (WT) and _Tg(foxd3:mCherry)_[ct110] (ct110) mutant larvae (representative of $n = 15$ biologically independent larvae per groups). **g** Blastemal cell proliferation in heterozygotes (WT) and homozygotes (ct110) _Tg(foxd3:mCherry)_[ct110] larvae at 24 hpA was detected using an anti-PH3 antibody (graph represents the mean number of positive cells ± SEM, $n = 21$ (for the WT groups) and $n = 18$ (for mutants) larvae from 3 independent experiments, one-tailed Mann-Whitney test was performed, $p = 0.0002$, ***$p < 0.001$). (**h**) Quantification of fin fold growth at 72hpA in WT and _Tg(foxd3:mCherry)_[ct110] mutant 6 dpf larvae (graph represents the mean ± SEM, $n = 22$ (for the WT group), $n = 19$ (for the mutant group) biologically independent larvae, one-tailed Mann Whitney test was performed, $p = 0.0001$, ***$p < 0.001$).

24 hpA (Fig. 3f). Altogether, these data revealed that NCdC are required for successful caudal fin fold regeneration, partly by inducing blastema cell proliferation and thereby blastema formation.

**_Foxd3_+ NCdC induce macrophage recruitment and activation during regeneration.** Macrophage subpopulations are essential for appendage regeneration in zebrafish[11] and the balance between M1- and M2-like macrophages is crucial for providing

the tightly regulated TNFα signal and inducing regeneration[12]. In Wallerian degeneration, SC, NCdC exert several functions allowing nerve regeneration, release some factors essential for macrophage recruitment. Particularly, it has been suggested that SC promote the recruitment of pro-inflammatory macrophages and induce their polarization toward an anti-inflammatory phenotype[23]. Therefore, we investigated the role of the _foxd3_+NCdC in macrophage recruitment and polarization during epimorphic regeneration. First, using _Tg(mpeg1:mCherry-_

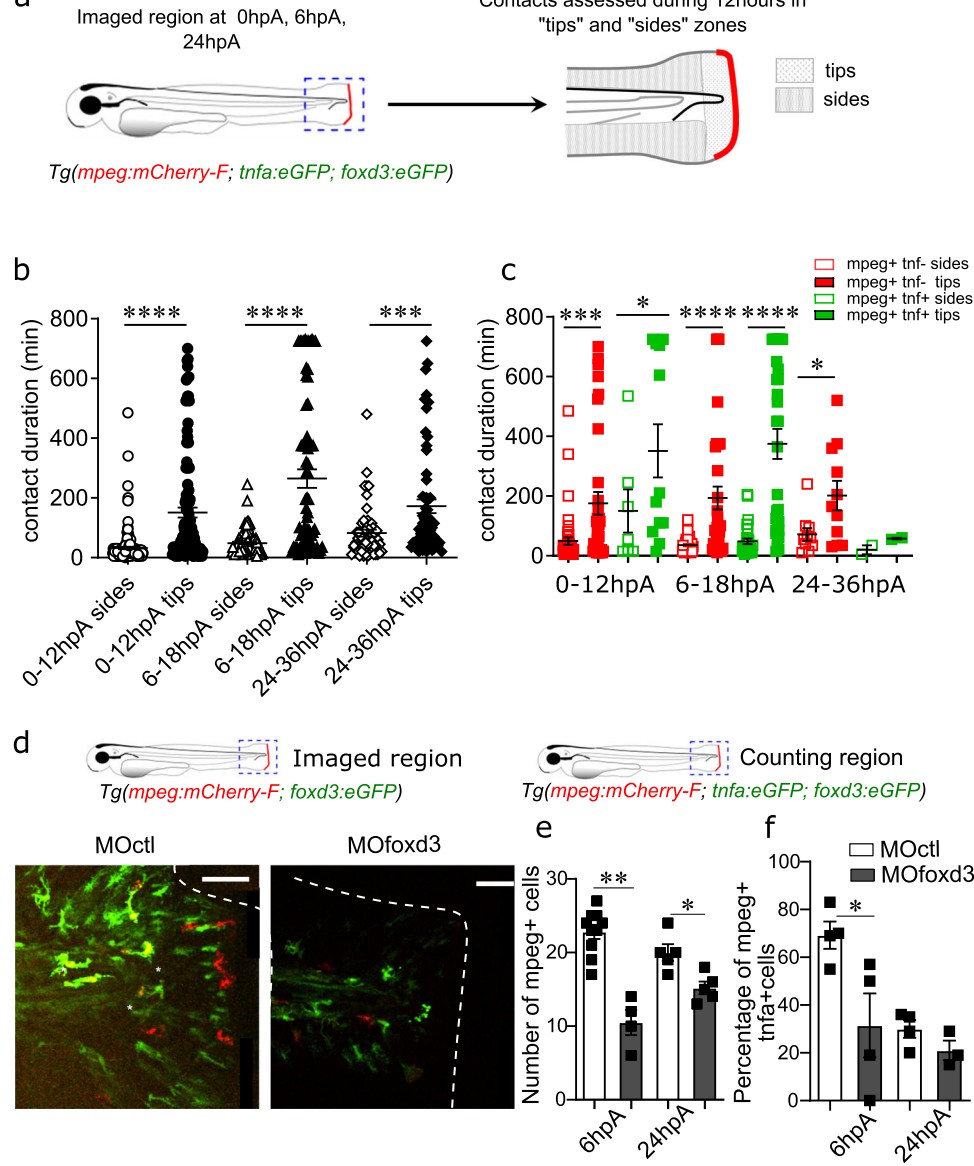

**Fig. 4 Foxd3+ NCdC interact with and promote the recruitment and activation of macrophages in the blastema. a** Schematic representation of the tip and sides where macrophages and *foxd3*+ NCdC contacts were assessed for 12 h by live confocal microscopy at 0hpA, 6 hpA, 24 hpA in *Tg(mpeg:mCherry-F; tnfa:eGFP-F; foxd3:eGFP-F)* larvae. **b** Quantification of the duration of macrophages and *foxd3*+NCdC contacts in the tips and sides during 12 h in *Tg(mpeg:mCherry-F; tnfa:eGFP-F; foxd3:eGFP-F)* larvae, at 0hpA ($n = 192$ for the sides group, $n = 124$ for the tips group), 6 hpA ($n = 105$ for the sides group, $n = 71$ for the tips group), and 24 hpA ($n = 54$ for the sides group, $n = 60$ for the tips group) (mean number of contact durations ± SEM, from at least $n = 5$ biologically independent larvae per time points, Kruskal Wallis test with Dunn's test for multiple comparison were performed, ****$p < 0.0001$,). **c** Quantification of the contact duration between macrophages (*mpeg* + and *tnfa*+) and *foxd3*+ cells in tips and sides during 12 h in the transgenic larvae, at 0 hpA, 6 hpA, 24 hpA (mean number of contact durations ± SEM, Kruskal-Wallis test and Dunn's test for multiple comparison were performed, *$p < 0.05$, ***$p < 0.001$, ****$p < 0.0001$, $n = 3$ biologically independent larvae per time point). **d** Fluorescence microscopy images of 3 dpf *Tg(mpeg1:mCherry-F/ tnfa:eGFP-F/foxd3:eGFP-F)* larvae injected with MOctl or MOfoxd3 (Scales bars = 50 μm). **e** Number of macrophages recruited at the wound site at 6 hpA and 24 hpA in MOctl- and MOfoxd3-injected *Tg(mpeg1:mCherry-F)* larvae (graph represents mean number of positive cells ± SEM, $n = 10$ biologically independent larvae for the control group at 6 hpA, $n = 4$ larvae for the MOfoxd3 group at 6 hpA, $n = 5$ larvae for both control and morphant group at 24hpA, one-tailed Mann-Whitney test was performed, $p = 0.0028$ for the 6 hpA time point, $p = 0.0106$ for the 24 hpA time point, *$p < 0.05$, **$p < 0.01$). **f** Percentage of pro-inflammatory macrophages at the wound site at 6 hpA and 24 hpA (graph represents mean number of *mpeg*+*tnfa*+ cells ± SEM, $n = 4$ biologically independent larvae per groups, one-tailed Mann-Whitney test was performed, $p = 0.0286$ for the 6 hpA time point, *$p < 0.05$).

F/tnfa:eGFP-F/foxd3:eGFP-F) 3 dpf larvae, we monitored the interactions between *foxd3*+NCdC and macrophage subpopulations at different time points during regeneration (0, 6, 24 hpA) using confocal microscopy (12 hours of live imaging), and quantified the interactions between these cell types in two different regions of the fin fold (tip and sides) (Fig. 4a). We found that macrophages and *foxd3*+NCdC had many contacts, and that

long-lasting contacts occurred preferentially in the tip region (Fig. 4b). Moreover, the long-lasting contacts in the fin fold tips were more frequent between *foxd3*+NCdC and *tnfa*+ macrophages (Fig. 4c, Movie 2). This result showed that contact frequency and duration between macrophages and *foxd3*+NCdC depend on the region of the regenerating blastema. The finding that long-lasting contacts were preferentially made by pro-inflammatory

($mpeg^+tnfa^+$) macrophages suggests a possible role of $foxd3^+$ NCdC in their activation/polarization. To address this hypothesis, we injected MO$foxd3$ in $Tg(mpeg1:mCherry\text{-}F/tnfa:eGFP\text{-}F/foxd3:eGFP\text{-}F)$ embryos. Confocal microscopy showed the massive decrease of $foxd3^+$NCdC concomitantly with a substantial reduction of macrophage frequency in the regenerating blastema of MO$foxd3$ morphants (MO$foxd3$) compared with control morphants (MO$ctl$) (Fig. 4d, e). Similarly, we did not observe any increase in $mpeg1$ expression in $Tg(foxd3:mCherry)^{ct110}$ mutants following fin fold ablation in contrast to wild-type larvae in which $mpeg1$ expression was significantly increased in the 24 hpA regenerating fin fold (Supplementary Fig. 4a). Of note, confocal microscopy and FACS analysis showed that the overall number of macrophages was not changed in $foxd3$ morphants compared with controls (Supplementary Fig. 4b-e). Conversely, the number of $tnfa^+$ macrophages was reduced in $foxd3$ morphants compared with control morphants (Fig. 4f, Movies 3 and 4). These results indicated that $foxd3$ silencing significantly impaired macrophage recruitment at the injured site and their activation toward a pro-inflammatory $tnfa^+$ phenotype. These observations are in accordance with our results showing that $foxd3^+$NCdC establish preferential contacts with pro-inflammatory $tnfa^+$ macrophages during regeneration, and strongly suggest a positive role for $foxd3^+$NCdC in macrophage recruitment and activation during caudal fin fold regeneration via a paracrine mechanism.

**Nrg1 and ErbB family members are expressed in blastemal cells and are required for appendage regeneration.** To identify the paracrine mechanism underlying $foxd3^+$NCdC role in the macrophage response during regeneration, we focused our attention on neuregulin 1 (NRG1), a critical factor for the development of NC cells and of some NCdC, including SC[32] that promotes the proliferation of damaged tissue cells in various models and during regeneration[33–35]. To determine whether NRG1/ErbB was one of signalling pathways that promote blastemal cell proliferation and macrophage pro-regenerative response, we first assessed the expression level of $nrg1$ and $erbb$ family members in our model. In situ hybridization analysis of 4 dpf larvae showed that $nrg1$ was upregulated in the regenerating blastema in larvae at 24 hpA compared with the intact caudal fin fold of control larvae (Fig. 5a). Then, analysis of the expression profile of the $nrg1$ splicing variants described in zebrafish (i.e., $nrg1.001$, $nrg1.002$, $nrg1.003$ and $nrg1.004$; ZFIN.org) revealed that $nrg1.001$ and $nrg1.002$ transcripts were not detectable in both intact and amputated fin fold (at 24 hpA), whereas the expression of $nrg1.003$ ($p = 0.0571$) and $nrg1.004$ ($p < 0.05$ $nrg1$) was increased (Fig. 5f). Next, we assessed the expression profile of $erbb2$ and $erbb3$ that have been described as critical factors for regeneration in zebrafish[36]. While $erbb3$ expression level was comparable in regenerating blastema at 24 hpA and in controls (supplementary Fig. 5a), $erbb2$ was significantly increased in response to caudal fin fold amputation (Fig. 5g). Then, we investigated the role of the NRG1 signalling pathway by adding PD168393 and AG1478, two specific inhibitors of the NRG1/ErbB pathway[36,37], to the water of 3 dpf larvae on amputation day and every day until 72 hpA (Fig. 5b). Compared with DMSO-treated amputated larvae (controls) in which caudal fin fold was fully regenerated at 72 hpA, in PD168393 or AG1478-treated amputated larvae fin fold regeneration was significantly impaired (Fig. 5d) and blastemal cell proliferation at 24 hpA was significantly reduced (Fig. 5c). These results are in line with previous studies and show that NRG1/ErbB signalling is necessary for caudal fin fold regeneration. Then, to determine which blastema cells expressed $nrg1$, we focused on NCdC and performed in situ hybridization in control and $foxd3$ morphants at 24 hpA. $Foxd3$

deficiency was associated with massive $nrg1$ downregulation (Fig. 5e). Of note, the four zebrafish $nrg1$ variants were not differentially expressed in response to amputation in $Tg(foxd3:mCherry)^{ct110}$ mutant larvae (Fig. 5f-g). Moreover, $nrg1.004$ expression levels were comparable in non-amputated $Tg(foxd3:mCherry)^{ct110}$ and wild-type larvae. In $Tg(foxd3:mCherry)^{ct110}$ mutants, $erbb2$, and $erbb3$ expression levels were not different compared with those in WT larvae and in response to amputation (Fig. 5g and Supplementary Fig. 5a). These results suggest that the significant $erbb2$ and $nrg1.004$ upregulation observed in wild-type larvae upon amputation depends on $foxd3$, and that $foxd3$-dependent NRG1/ErbB2 signalling has an essential role in appendage regeneration, by highlighting the correlation between $nrg1$ expression and presence of $foxd3^+$ cells in the regenerating blastema.

**The Nrg1/ErbB2 signalling pathway is part of the molecular dialogue between macrophages and $foxd3^+$ neural crest-derived cells during regeneration.** As $foxd3$ has a role in NRG1/ErbB2 signalling during regeneration and mammalian macrophages express several NRG1 receptors and migrate in response to NRG1 in vitro[38], we investigated whether $foxd3^+$NCdC promoted macrophage recruitment and activation through the NRG1/ErbB2 signalling pathway during caudal fin fold regeneration. To this aim, we first treated 3 dpf amputated zebrafish with AG1478, a specific ErbB inhibitor (Fig. 5h). In treated zebrafish larvae, the number of macrophages ($mpeg^+$ cells) was significantly decreased in the injured fin fold at 6 and 24 hpA (Fig. 5i). Moreover, macrophage activation and polarization toward a pro-inflammatory phenotype were reduced at 6 hpA, as indicated by the lower number of $tnfa^+$ macrophages in the regenerating fin fold of treated larvae compared with DMSO controls (Fig. 5j). These results suggest that NRG1/ErbB2 is necessary for macrophage recruitment and polarization during caudal fin fold regeneration.

As macrophage response to NRG1 has never been described during zebrafish caudal fin fold regeneration, we assessed the expression profile of $nrg1.004$, $erbb2$, and $erbb3$ in FACS-sorted macrophage populations (Fig. 6a) from 3 dpf $Tg(mpeg1:mCherry\text{-}F/tnfa:eGFP\text{-}F)$ zebrafish larvae at 6 hpA and 24 hpA. Although $nrg1.004$, $erbb2$, $erbb3$ were expressed in all sorted cell populations (Fig. 6b, c, d, e, f, g), $nrg1.004$ transcript level was particularly abundant in negative cells (i.e., non-macrophage cells) derived from amputated larvae at 6 hpA and 24 hpA, compared with control. $Erbb2$ and $erbb3$ also were mostly expressed in negative cells, but in macrophage populations, $erbb2$ and $erbb3$ (to a lower extent) transcripts were enriched in $tnfa^+$ macrophages at 6 hpA, and also at 24 hpA. These data show that zebrafish macrophages express $nrg1$, $erbb2$, and $erbb3$, and that $erbb2$ and $erbb3$ are upregulated in the $tnfa^+$ pro-inflammatory macrophage subset upon caudal fin fold amputation. This specific expression in $tnfa^+$ macrophages and their preferential long-lasting contacts with $foxd3^+$NCdC (Fig. 4) strongly suggest that macrophage recruitment and polarization are regulated by $foxd3^+$ NCdC through the NRG1/ErbB signalling pathway.

**Discussion**
In this study, we provide a cell atlas of the regenerating zebrafish caudal fin fold using the high-dimensional scRNA-seq approach. We found that the regenerating zebrafish caudal fin fold comprises seven cell types and identified a cell population within the regenerating fin fold: $foxd3^+$ NCdC. We then demonstrated that $foxd3^+$ NCdC are required for caudal fin fold regeneration and that they regulate macrophage recruitment and activation, a

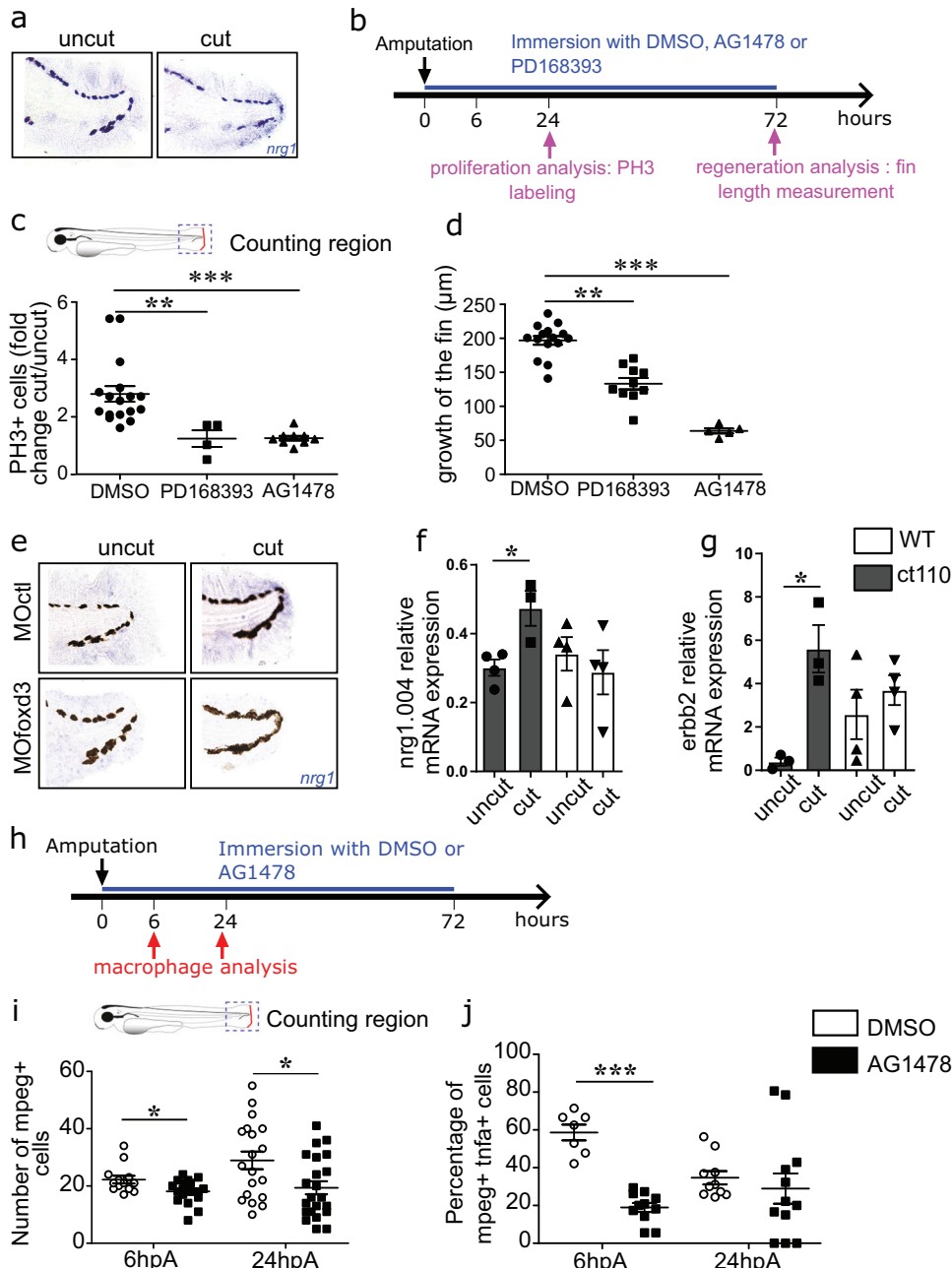

**Fig. 5 Nrg1 and ErbB family members are expressed in blastemal cells and are required for appendage regeneration. a** Representative image of *nrg1* mRNA expression in uncut caudal fin fold and at 24hpA (from *n* = 15 biologically independent larvae). **b** Wild type 3dpf larvae were amputated or not and exposed to DMSO, AG1478, or PD168393 for 72 h. Cell proliferation and fin fold growth were measured at 24hpA and 72hpA, respectively. **c** Quantification of cell proliferation in 24 hpA larvae after the indicated treatments (graph represents means ± SEM, *n* = 17 larvae for DMSO group, *n* = 4 larvae for PD168393 group, *n* = 9 larvae for AG1478 group, Kruskal-Wallis test and Dunn's test for multiple comparisons were performed, **$p$ < 0.01, ***$p$ < 0.001). **d** Quantification of fin fold growth in 72 hpA wild type larvae after the indicated treatments (error bars show the SEM, *n* = 15 larvae for DMSO group, *n* = 10 larvae for PD168393 group, *n* = 5 larvae for AG1478 group, Kruskal-Wallis test and Dunn's test for multiple comparisons were performed **$p$ < 0.01, ***$p$ < 0.001). **e** *nrg1* expression by in situ hybridization analysis in intact caudal fin folds and at 24hpA of MO*foxd3* or MO*ctl* larvae (from *n* = 15 biologically independent larvae). **f** Relative expression of *nrg1.004*, and **g** *erbb2* mRNA in wild type (WT) and *Tg(foxd3:mCherry)*[ct110] mutant (ct110) larvae at 24hpA was assessed by RT-PCR using *ef1a* as reference gene (data are the mean ± SEM, *n* = 15 larvae per groups from 3 independent experiments, one-tailed Mann–Whitney test was performed, **f** $p$ = 0.0286, **g** $p$ = 0.05, *$p$ < 0.05). **h** 3dpf *Tg(mpeg1:mCherry-F/tnfa:eGFP-F)* larvae were amputated or not and exposed to DMSO or AG1478. Macrophage recruitment was analysed at 6 hpA and 24 hpA. **i** Macrophage recruitment at the wound site in AG1478 or DMSO-treated *Tg(mpeg1:mCherry-F)* larvae at 6 hpA and 24 hpA (graph represents mean number of mCherry-positive cells ± SEM, *n* = 13 at 6 hpA and *n* = 24 at 24hpA for DMSO treated groups, *n* = 19 at 6 hpA and *n* = 22 at 24 hpA for AG1478 treated groups, one-tailed Mann–Whitney test was performed, $p$ = 0.0154 for the 6 hpA time point, $p$ = 0.0143 for the 24 hpA time point, *$p$ < 0.05). **j** Macrophage polarization at the wound site in AG1478 or DMSO-treated *Tg(mpeg1:mCherry-F/tnfa:eGFP-F)* larvae at 6 hpA and 24 hpA (graph represents mean number of mCherry⁺ eGFP⁺ cells ± SEM, *n* = 7 at 6 hpA and *n* = 10 at 24 hpA for DMSO treated groups, *n* = 11 at 6 hpA and *n* = 12 at 24 hpA for AG1478 treated groups, one-tailed Mann–Whitney test was performed, $p$ = 0.0003 for the 6 hpA time point, ***$p$ < 0.001).

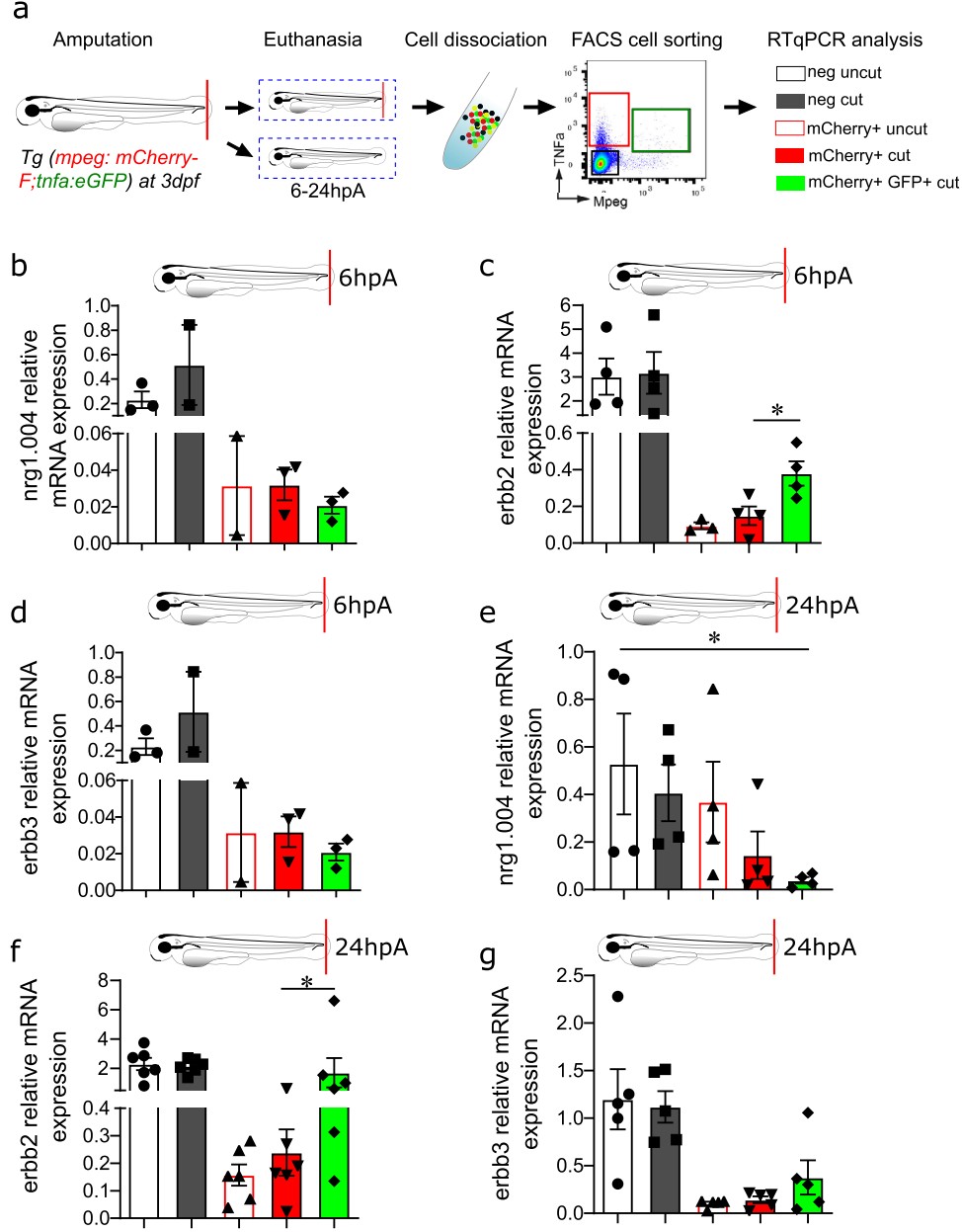

**Fig. 6 *Erbb2* is expressed by all blastemal cells including macrophages, and particularly pro-inflammatory macrophages. a** *Tg(mpeg1:mCherry-F/ tnfa:eGFP-F)* larvae were amputated or not (control) at 3 dpf, and cells were dissociated at 6 or 24 hpA and sorted by FACS. Red, green, and black gates represent *mCherry+*, *mCherry+eGFP+*, and *mCherry−eGFP−* cell populations, respectively. Relative expression of **b, e** *nrg1*, **c, f** *erbb2*, and **d, g** *erbb3* in *mCherry −eGFP−*, *mCherry+* and *mCherry+eGFP+* cells at 6 hpA **b, c, d** and 24 hpA **e, f, g** was quantified by RT-PCR on separated cells using *ef1a* as reference gene. **b, c, d** Graphs represent the mean value ± SEM, n = 200-300 pooled larvae from 4 independent experiments; one-tailed Mann-Whitney test was performed, $p = 0.0286$, *$p < 0.05$. **e, f, g** Graphs represent the mean value ± SEM, $n = 200$–300 pooled larvae from 6 independent experiments; one-tailed Mann-Whitney test was performed, **e** $p = 0.0143$, **f** $p = 0.0465$, *$p < 0.05$.

pivotal step in the regeneration process[12], through the NRG1/ ErbB signalling pathway.

Our study provides a robust and efficient methodology to unbiasedly demonstrate the heterogeneity of the blastema formed after caudal fin fold amputation in zebrafish larvae. Live clonal analysis in zebrafish has allowed visualizing and quantifying, at high resolution, blastema formation, revealing the plasticity of blastema cells during regeneration. However, the exact timing of blastema formation and its heterogeneity were so far undefined. Here, we identified all the different cell types that compose the blastema, and also an orchestrator cell that governs the regeneration process. Indeed, our results show that *foxd3+* NCdC drive

the continuous regeneration of the injured tissue. Blastema formation, which is required for regeneration, is mediated through the release of trophic factors by NCdC in newts and mammals. For instance, production of the mitogenic Anterior Gradient protein (nAG) first by NCdC and then by the AEC is necessary and sufficient for blastemal cell proliferation and the rescue of the regeneration potential of denervated limbs[18]. Similarly, a recent study in mammals demonstrated that nerve-associated SC precursors dedifferentiate and secrete growth factors that promote blastema expansion and digit regeneration. Here, by combining the use of 4D confocal microscopy and transgenic larvae to track both *foxd3+* NCdC and *rcn3+* mesenchymal cells in the

regenerating caudal fin fold from 6 hpA, we showed that all proliferating cells are mesenchymal cells that proliferate in contact with foxd3[+] NCdC. This observation highlights NCdC key role in blastema expansion and strongly suggests that blastemal cell proliferation is NCdC-dependent. This dependence was confirmed by the finding that Tg(foxd3:mCherry)[ct110] mutants and foxd3 morphants cannot regenerate their fin fold after amputation. Altogether, these studies demonstrated that in regenerative species, appendage regeneration is possible through a conserved mechanism that relies on NCdC presence and activation. Of note, NCdC and other cell types that play an important role during appendage regeneration, such as neurons and endothelial cells, were not identified in the recent published study that described the cellular diversity of postinjury adult zebrafish fin[10]. This might be due to a systematic bias against some discrete cell subsets that might be generated during sample preparation or in the droplet-based scRNA-seq system. Increasing the number of cells or performing scRNA-seq of foxd3[+]-sorted cells within the regenerating fin fold would help to identify this discrete cell subset and better study its role in the regeneration process of adult zebrafish fin.

The tight and long-lasting contacts between macrophages and foxd3[+] NCdC suggested a functional interplay between these cell types, leading to the polarization toward a pro-inflammatory phenotype of the macrophages that interact with NCdC. Moreover, foxd3 silencing did not affect the total number of mpeg1[+] macrophages in the entire zebrafish, but it significantly impaired their recruitment and activation at the injured site. Therefore, foxd3[+] NCdC are required for the generation of pro-inflammatory macrophages, presumably through direct contact, leading to the production of TNFα, which is critical for blastema formation and priming of caudal fin regeneration in zebrafish[12]. These findings indicate that NCdC could promote the proliferation of blastemal cells indirectly through macrophage activation.

We also observed a substantial upregulation of nrg1 in the regenerating blastema. NRG1 is a critical factor for the development of NC cells and some NCdC, including SC[32], that promote the proliferation of damaged tissue cells in various models and during regeneration[33,34]. In zebrafish and mouse, NRG1 promotes and stimulates cardiomyocyte proliferation during heart regeneration and repair, respectively[35,39] in an ERBB2-dependent manner[40,41]. ErbB2-ErbB3 inhibitors significantly decrease progenitor cell proliferation in the blastema[36]. Nrg1/ErbB2 signalling might play an important role in cardiomyocyte proliferation in the regenerating heart of zebrafish by inducing their metabolic reprogramming[42]. In mammals, NRG1 controls SC proliferation and migration through binding to ErbB2-ErbB3[43]. Moreover, during cardiac regeneration in adult mice, ERBB2 overexpression in cardiomyocytes promotes an epithelial-mesenchymal-like regenerative response characterized by remodelling of their cytoskeleton, junction dissolution, migration, and extracellular matrix replacement[44]. In line with these studies, we found that erbb2 was upregulated concomitantly with the increased expression of nrg1 in the regenerating blastema. Moreover, exposure to specific ErbB inhibitors significantly impaired blastemal cell proliferation and caudal fin fold regeneration. The finding that nrg1 and erbb2 upregulation in response to fin fold amputation is foxd3-dependent highlights the correlation between nrg1 and the presence of foxd3[+] cells in the regenerating blastema. This suggests that NCdC are required for regeneration through the secretion of NRG1, a paracrine and mitogenic factor. Our data are reminiscent of mammalian digit tip regeneration, where SC precursors secrete paracrine factors, such as oncostatin M (OSM) and PDGF-AA, that enhance mesenchymal cell proliferation and regeneration[19]. Moreover, during mammalian cardiac regeneration, OSM and NRG1 induce cardiomyocyte proliferation and dedifferentiation both in vitro and in vivo[39,45,46].

As expected from our previous work, we identified macrophages that exert a critical role in the zebrafish caudal fin fold regeneration[12] among a subgroup of mpeg1[+] cells in intact and to a greater extent in regenerating fin fold tissues. We then discovered that zebrafish macrophages express erbb2 and erbb3, like mammalian macrophages that express erbb2, erbb3 and migrate in response NRG1 in vitro[38]. These receptors are expressed particularly by tnfa[+] pro-inflammatory macrophages. Using an ErbB specific inhibitor, we observed a significant decrease in the number of macrophages in the injured fin fold at 6 and 24 hpA associated with a significant reduction of macrophage activation and polarization toward the pro-inflammatory phenotype. This result further confirms NCdC pivotal role during regeneration not only as cells that express mitogenic factors, such as NRG1, but also as cells that induce the release of TNFα by activated pro-inflammatory macrophages. In the context of nerve injury, NCdC releases cytokines essential for M1-like macrophage recruitment and their polarization toward an anti-inflammatory M2-like macrophage[23]. Moreover, the sequential and well-coordinated recruitment of these two macrophage subsets during appendage regeneration provides the tightly regulated TNFα signal that orchestrates the process[12]. Consistent with studies describing erbb2, erbb3, and erbb4 expression in macrophages[38], we elucidated the functional dialogue between macrophage and NCdC during epimorphic regeneration in zebrafish. However, and in contradiction with our results, it was previously shown that the ErbB/NRG1 signalling pathway is not required for immune cell recruitment to the wound in zebrafish caudal fin regeneration[36]. This discrepancy could be explained by the use in this previous study of a transgenic line that is not specific for macrophages or myeloid cells, and by the absence of quantification of the observed effect.

It was previously shown that during zebrafish regeneration, the ErbB/NRG1 signalling pathway[36] has a role in blastema mesenchymal cell proliferation and migration. However, these authors did not identify the origin of NRG1-producing cells. As the NRG1/ErbB signalling pathway is involved in SC precursor development and biology, the authors used the colourless/sox10 mutant in which the development of NC derivatives, particularly SC, is altered to study their role in caudal fin regeneration. They did not observe any regeneration impairment in this mutant and concluded that SC are not required for regeneration[36]. This discrepancy with our results could be explained by the fact that foxd3 exhibits a large but incomplete spatiotemporal overlap with sox10 expression[47]. Moreover, the spatiotemporal distribution of the fluorescent proteins in three transgenic lineages, Tg(foxd3:GFP), Tg(sox10:eGFP), and Tg(sox10:mRFP), shows major differences during early NC development[36]. Thus, zebrafish transgenic lines are powerful experimental tools for cell lineage tracing investigations, however, their characterization is critical to address a specific question and provide accurate conclusions. Here, on the basis of previous studies and our results showing the presence of foxd3[+] cells but not of sox10[+] cells in the caudal fin fold mesenchyme, we focused our attention on foxd3[+] cells to study the role of NC cell derivatives during epimorphic regeneration of the caudal fin fold. Moreover, the transgenic line Tg(foxd3:GFP) has been used in several studies on NC derivatives, including SC, that also demonstrated the pertinence of focusing on foxd3 to address NCdC role in appendage regeneration[48,49].

In addition to foxd3[+] cells within the regenerating caudal fin fold, we noticed phenotypic and morphological changes of rcn3[+] caudal fin fold mesenchymal cells that acquired NC-like properties. In line with this result, we observed that while the frequency of eGFP[+]foxd3[+] NCdC did not change during the first 24 h of the regeneration process, the frequency of mCherry[+] and mCherry[+]eGFP[+] mesenchymal cells was significantly increased

in the regenerating caudal fin fold at 24 hPA compared with the intact fin fold at the same developmental stage. It will be interesting to determine the identity of these mesenchymal cells that express NC markers and their exact function during regeneration. There is a controversy about the origin of mesenchymal cells in the caudal fin in teleost fishes. Kague and colleagues proposed the NC origin of the caudal fin using genetic-based lineage tracing[50]. Conversely, Lee and colleagues argued that fin mesenchymal cells derive entirely from the mesoderm without any NC contribution[51]. However, this transition from NCdC toward mesenchymal cells specifically in the regenerating tissue echoes what was observed in the context of mouse digit tip regeneration[52]. Indeed, single-cell profiling showed that mesenchymal blastemal cells are distinct from control digit mesenchymal cells[52]. The authors concluded that within the blastema, cells acquire a mesenchymal transcriptional state and participate in the regeneration of dermis and bone[52]. Therefore, we could extrapolate that within the blastema formed after caudal fin fold amputation, NCdC adopt a mesenchymal cell phenotype to contribute to the caudal fin mesenchymal tissue regeneration.

Altogether, these findings demonstrate that $foxd3^+$ NCdC are one of the cell subsets in the regenerating blastema and that they play a major role in epimorphic regeneration in zebrafish. NCdC are required for blastema cell proliferation and blastema formation through the release of the trophic factor NRG1 and the activation of $erbb2^+$ and $erbb3^+$ blastemal macrophages. Activated macrophages express TNFα and thus provide the accurate signal to prime regeneration in zebrafish. This study proposes an integrated view of the regenerative process in a vertebrate in which NCdC activates macrophages and secrete mitogenic paracrine factors. Understanding how the expression of such factors is regulated could be a key to activate lost regeneration processes or to improve the healing response in mammals. Finally, this study underlies the crucial importance of investigating the tight crosstalk between cells during this fine regulated process, which could lead to appropriate therapies for regenerative medicine.

## Methods

**Ethics statement**. All animal experiments described in this study were carried out at the University of Montpellier according to the European Union guidelines for the handling of laboratory animals (http://ec.europa.eu/environment/chemicals/lab_animals/home_en.htm) and were approved by the (http://ec.europa.eu/environment/chemicals/lab_animals/home_en.htm) and were approved by the Comité d'Ethique pour l'Expérimentation Animale under reference CEEA-LR- B4-172-37 and APAFIS#5737-2016061511212601 v3.

**Zebrafish lines and maintenance**. Embryos were generated from pairs of adult fish by natural spawning and raised in tank water at 28.5 °C[53]. Experiments were performed using the AB zebrafish stain (ZIRC), and the transgenic line $Tg(mpeg1:mCherry-F)$ to visualize macrophages[54], $Tg(tnfa:eGFP-F)$ to visualize $tnfa$ expression[17], $Tg(rcn3:gal4/UAS:DsRed)$ to visualize mesenchymal cells[26], $Tg(foxd3:eGFP-F)$[55] and $Tg(sox10:eGFP-F)$[56] to visualize NC cells, $Tg(col2a:mCherry)$ to visualize chondrocytes. Homozygous larvae from the $Tg(foxd3:mCherry)^{ct110}$ line were used as $Foxd3$ mutants[30]. Embryos were obtained from adult fish pairs by natural spawning and were raised at 28.5 °C in tank water.

**Larva manipulation for regeneration assays and imaging**. Caudal fin fold amputation was performed in 3 dpf larvae under anaesthesia with 0.016% Tricaine (MS222, Sigma) in zebrafish water using a sterile scalpel[57]. For imaging, live embryos were anesthetized in 0.016% Tricaine, and positioned in 35 mm glass-bottom dishes (FluoroDish™, World Precision Instruments). They were mounted in 1% low melting point agarose (Sigma) with Tricaine. Light microscopy was performed using an MVX10 Olympus macroscope equipped with an MVPLAPO 1X objective and XC50 camera. Z-stacks series were obtained using an inverted confocal microscope Leica TCS SP5 (Leica Application Suite V3.2) and TCS SP8 (Leica Application Suite V3.5) equipped with an HCXPL APO 40x/1.25-0.75 oil and an HC PL APO 0.70 ∞ (infinity) 20x objective (Leica). The mCherry signal was excited with a 560 nm laser, and GFP with a 490 nm laser. Datasets were analysed using Fiji Software (ImageJ 1.52p)[58].

**Cell isolation, library preparation and data processing of 10x Genomics Chromium scRNA-seq data**. Approximately, 150 cut and uncut caudal samples were collected and dissociated into a single-cell suspension. Cell viability and aggregation were tested prior to proceeding with the 10X Genomics protocol for 3' transcript capture and single-cell library preparation. The concentration of freshly dissociated cells was adjusted to 700–800 cells/μl in PBS aiming to capture 4000 cells by the 10X Genomics device. Briefly, the manufacturer's protocol (Chromium™ Single Cell 3' Reagent kit v3.1) was followed to prepare single-cell libraries for Illumina sequencing. Libraries quantification was performed using the Fragment Analyzer system (NGS High Sensitivity Kit) and qPCR (ROCHE, Light Cycler 480). Sequencing was performed in paired-end mode with an S1 flow cell (28/8/87 cycles) using a NovaSeq 6000 sequencer (Illumina) at the MGX core facility of Montpellier. We used cell ranger mkfastq and cellranger count pipelines from the Cell Ranger Single Cell software by 10x Genomics (http://10xgenomics.com) for the initial quality control, sample demultiplexing, mapping, and quantification of raw sequencing data. Quality controls for scRNA-seq are provided (Supplementary Data 5). Raw scRNA-seq data were processed using the Cell Ranger software (Cell Ranger v3.1.0) provided by 10X Genomics with the default options. Briefly, files were first converted to the fastq format, and then sequences were aligned to the $Danio$ $rerio$ reference genome (danRer11) to generate single-cell feature counts using the standard 10x Genomics CellRanger Count pipeline with default parameters. The Cloud software (version 3.1.1) was used to visualize and analyze the results obtained with Cell Ranger. The Uniform Manifold Approximation and Projection (UMAP) dimensional reduction technique was used to visualize data. Output graphing allowed the visualization of cell cluster identity and marker gene expression.

**Morpholino injections, Drug treatments**. For $foxd3$ (NM_131290) knock down experiments, morpholino antisense oligonucleotides (Gene Tools) against the ATG site were used ($MOfoxd3$): 5' TGCTGCTGGAGCAACCCAAGGTAAG 3'. As a control, a Control morpholino ($MOctl$) from Gene Tools was used: 5' AATCAC AAGCAGTGCAAGCATGATG 3'. Two-three nl of each morpholino at 500 μM concentration was injected in one-cell stage embryos with a Femto. Jet from Eppendorf. No side effect was observed. For Nrg1 signalling inhibition, AG1478 (Sigma), and PD168393 (Sigma) were diluted directly in fish water at 10 μM, and the treatment was renewed every 24 h after amputation.

**In situ hybridization**. The plasmid containing $junbl$ (PCRII-junb-l) was kindly sent by Atsushi Kawakami (Department of Biological Information, Tokyo Institute of Technology, Japan), and the plasmid containing $nrg1$ was kindly sent by Kenneth D. Poss (Department of Cell Biology, Howard Hughes Medical Institute, Duke University Medical Center, Durham, United States). Digoxigenin (DIG)-labelled (Roche) sense and anti-sense RNA probes were prepared using the in vitro Transcription kit (Biolabs). In situ hybridization of whole embryos was performed as detailed in[59]. Embryos were imaged with an Axio Scan from Zeiss with a 40X/0.95 objective (Zeiss Axio Scan.Z1; Zeiss Axio Imager Z.2).

**Cell proliferation detection**. For quantification of cell proliferation, whole embryos were fixed in 4% paraformaldehyde overnight and stained using an anti-phosphorylated histone 3 antibody (Cell Signaling, ref 9701, dilution:1/500)[12].

**Larval tail RNA preparation and quantitative RT-PCR**. To determine the relative expression of $nrg1$ type III, type IV, $erbb2$, $erbb3$, $mpeg1$, $foxd3$, and $ef1a$, total RNA from larval fin fold (pools of 20 or 30 fin fold) was prepared at 24hPA. Total RNA (20 ng) was reverse-transcribed with the High-Capacity RNA Reverse Transcription kit (Applied Biosystems, France) RT-qPCR analyses were performed using the Light Cycler480 system and the following primers: ef1a.5(5'-TTCTGTTACCTGG CAAAGGG-3'), ef1a.3(5'-TTCAGTTTGTCCAACACCCA-3'), erbb2.5(5'-CCATG GCACGGGATCCCTCA-3'), erbb2.3(5'-GCTGTTGCGCCCACAGGAAG-3'), erb b3.5(5'-GCCCGTGGAGCTCAGAGCATT-3'), erbb3.3(5'-CCAACGGGAAAGGC GCTACTG-3'), nrg1.003.5 (5'-GGCCAGCTTCTACAAAGCTGAGGA-3'), nrg1.0 03.3 (5'-GCTGCAGCGGTTTCGCTCTCG-3'), nrg1.004.5 (5'-TGGGATTGAATT TATGGAAGCTGAGGA-3'), nrg1.004.3 (5'-GGTGGAGGGTGAGGGTGTTG-3'), foxd3.5 (5'-CCGGGAGAAGTTTCCGGCCT-3'), foxd3.3 (5'-TGGGGGTCGAGG GTCCAGTA-3'), mpeg.5 (5'-GTGAAAGAGGGTTCTGTTACA-3'), and mpeg.3 (5'-GCCGTAATCAAGTACGAGTT-3').

**Monitoring fin fold regeneration, macrophage subset count, cell proliferation, morphological changes and statistical analysis**. Caudal fin fold regeneration was monitored by measuring the fin fold growth length from the transected plan (end of the notochord) up to the most proximal end of the fin fold with the Fiji software. Macrophages and cell proliferation in the wound region were measured directly on images acquired by microscopy using the indicated reporter lines and staining. Morphological changes of $foxd3^+rcn3^+$ cells were assessed with the Fiji software using the circularity and roundness plugins. These plugins are an extended version of the ImageJ Measure command that calculates object circularity using the formula: circularity = $4pi(area/perimeter^2)$. A circularity value of 1.0 indicates a perfect circle. Values approaching 0.0 indicate an increasingly elongated polygon. As roundness = $4area/(\pi major\_axis^2)$, roundness is more relative to the area of the object compared with the main axis. It could be considered as opposite to the elongation factor, whereas circularity refers to the object shape compared to a

perfect circle. Graphs show the mean ± standard error of the mean (SEM). The Mann–Whitney test was performed to test the significance of the data presented in all the figures except for Fig. 3a for which Kruskal–Wallis ANOVA with the Dunn's post-hoc test was performed to test the significance of the data using the GraphPad Prism 6 software (San Diego, CA, USA).

**Reporting summary**. Further information on research design is available in the Nature Research Reporting Summary linked to this article.

## Data availability

All datasets generated in this study have been deposited in the Gene Expression Omnibus repository under the series number GSE158851. Source Data are included with this paper. Source data are provided with this paper.

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

## Acknowledgements

This work was supported by Inserm and University of Montpellier. We thank the MRI facility for their assistance, T. Sauka-Spengler (University of Oxford, United Kingdom) for shipping the *Tg(foxd3-mcherry)^{ct110}* zebrafish line, M. Bagnat for shipping *Tg(rcn3:gal4/UAS:mCherry)* line. A. Kawakami (Tokyo Institute of Technology, Japan) for the *junbl* plasmid, K. Poss (Duke University Medical Center, USA) for the *nrg1* plasmid, R. Kelsh (University of Bath, United Kingdom) for the *Fkd6* in situ hybridization plasmid. MP and DS acknowledge financial support from the France Génomique National infrastructure, funded as part of "Investissement d'avenir" programme managed by the Agence Nationale pour la Recherche (contract ANR-10-INBS-09). We also thank the Zebrafish facility of the University of Montpellier, the MRI facility for their assistance and the CARTIGEN platform.

## Author contributions

B.L.B. and F.D. designed experiments with input from A.B., A.S., C.B., M.P., D.S., G.T., P.L.C., M.N.C., M.M., and C.J. B.L.B., A.B., C.B., G.T., P.L.C., M.N.C., and M.M. performed experiments, and B.L.B., A.S., M.P., D.S., and M.M. performed the analysis. F.D. and B.L.B. wrote the paper with input from S.A. and C.J.

## Competing interests

The authors declare no competing interests.

## Additional information

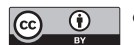

