## [Peer Review File · Nature Communications]

NRG1/ErbB signalling controls the dialogue between macrophages and neural crest-derived cells during zebrafish fin regenerationREVIEWER COMMENTS

Reviewer #1 (Remarks to the Author):

The study by Beryl et al., demonstrates that a population of neural crest-derived cells exist in the regenerating zebrafish caudal fin fold that are associated with macrophage recruitment, blastema cell proliferation, and NRG1/ErbB signaling. There is a growing number of examples in which neural crest-derived Schwann cells are mediators of regeneration including salamander limb regeneration and mouse digit regeneration. This paper adds to these compelling lines of evidence with yet another example of the importance of the neural crest during appendage regeneration. There have also been several studies demonstrating the importance of macrophages during appendage regeneration in salamanders, mice, and zebrafish, but the mechanism of their recruitment and exact role is still unclear. Lastly, NRG1 signaling has become a signaling pathway shown to be important for appendage and heart regeneration in multiple species. Again, the mechanism of action for NRG1/ErbB signaling is still a work in progress.

The study by Beryl et al., link all these separate lines of study together in the zebrafish dorsal fin fold model of appendage regeneration. They show that neural crest-derived cells recruited the proper macrophage phenotype necessary for regeneration and do so using the NRG1/ErbB2 signaling. Although the study does not answer all open questions in these links, it is an important step forward for explaining that these three apparently unlinked phenomena are linked with one another in order to coordinate appendage regeneration in mice, salamanders, and zebrafish.

I find the data compelling and the main conclusions supported by the experiments, but I do have some concerns with the particular cells that are being analyzed by the scRNAseq as well as the actual identity of the foxd3+ cells. Please find my comments and concerns below moving from beginning to end of the manuscript:

1) The reference numbering in the following sentence needs to be corrected: "transient accumulation of pro-inflammatory macrophages 178."

2) Figure 1a shows transgenic animal lines including a foxd3:eGFP reporter and an rcn3 mcherry reporter line, but these are not described in the first Results section. It is confusing to the reader because the presentation of the transgenic lines suggest that some sort of cell selection was performed, but this seems not to be the case in the written text. I am aware of the experimental design, but if the reader moves through the paper in a linear manner, they may be misled with Figure 1a if there is no description in the manuscript text. Furthermore, I know this isn't the case, but as it is presented in Figure 1a, it looks like foxd3 was already known to be a marker of an important cell type. I suggest removing the representation of these transgenic lines in figure 1 of the scRNAseq experimental design.

3) Concerning the following sentence: "We then assigned an identity to the cell clusters that composed the blastema of the regenerating fin fold on the basis of the key marker genes that were significantly enriched in each cluster." How do the authors know that the seven cell populations observed in the 24 hours sample are found in the blastema rather than uninjured tissues? When comparing Figure 1b with Supplemental Figure 1, they seem to have the same seven cell populations. Does this mean all seven populations are found in uninjured and blastema tissue with similar expression profiles? Or does it mean non-regenerating tissue was collected in the regenerating sample? I feel some in situ hybridization or IHC would help clear up this distinction, which is an important one when trying to understand appendage regeneration.

4) Based upon the following sentence, it is assumed that hmx3a and irg11 are up-regulated during

regeneration. I don't see evidence of this presented in this portion of the results. Furthermore, are similar cells found in the uninjured fin? "Cells in this cluster also strongly expressed *hmx3a* (a gene encoding a protein predicted to have sequence-specific DNA binding activity) and *irg1l* (a gene encoding a protein involved in the inflammatory response to wounding), suggesting that this cell population might specifically respond to amputation (Fig 1d-e)."

5) In Figure 1, please include which cell types are represented in each cluster either in Fig. 1B, C, or D. It will be informative to understand the relative contribution of each cell type to the total population.

6) It is unclear why clusters are determined "NCdC/mesenchymal cells and epidermis/mesenchymal cells". Is this because the clusters express markers of each cell type? This should be explained in the results where these cells are first described in a bit more detail.

7) I am surprised no melanophore cell signatures were found in the scRNAseq dataset. Are there no melanophores in the tail fin at this point of development?

8) Did the FACS data presented in Supplemental Figure 2a support the conclusion of increased mcherry/GFP+ cells at 24hpa presented in Figure 2i? I also think the error bars are missing in the uninjured control.

9) I think the authors meant to say "beneath" rather than "beyond" in the following sentence: "Caudal fin fold regeneration involves the critical step of blastemal cell proliferation in the area beyond the apical epithelial cap (AEC) from 6 hpa"

10) When I complete the Results section, I feel like the main conclusion I am missing is the identification of the NCdC cell type. Will these cells become Schwann cells in the adult, or do they generate pigment cells, or even mesenchyme? What is the normal function of these cells? It is most likely not to support regeneration.

Sincerely,

James Monaghan
Northeastern University

Reviewer #2 (Remarks to the Author):

In the manuscript, the authors identified a subset of blastemal cells with *Foxd3* positives which regulate macrophage recruitment and polarization via the *NRG1/ErbB* pathway. This is an interesting study and the authors provided solid evidence to support their finds. It is really a nice work.

Major comments:

(1) This study identified 7 clusters (cell subpopulations, Figure 1B and Supplementary Figure 1A). Are the M1/M2 macrophages part of these 7 clusters or the scRNA-seq data do not have such resolution to investigate macrophages?

(2) The only limitation of this study is that the fin regeneration/major conclusion had a context of early development (3 days post fertilization (dpf)). The fin regeneration process and mechanism can be different for adults (e.g., timing, gene expression patterns).

(3) Unsupervised UMAP analysis identified the same 7 cell clusters in both intact and amputated fins (Figure 1B and Supplementary Figure 1A). However, the cluster identify was based on several marker genes. It is unclear whether these cell clusters are really identical in both intact and amputated fins. The main problem of marker gene based methods to identify cell population (and align cell populations

in different conditions) is that the definition of marker genes are context dependent. There are many statistical methods which can compare cell subpopulations in different conditions. For example, the Seurat R package (<https://satijalab.org/seurat/v3.2/integration.html>) can perform a canonical correspondence analysis (CCA) to compare and visualize different subpopulations in different conditions. This can be helpful to compare these 7 cell clusters in intact and amputated fins.

Reviewer #4 (Remarks to the Author):

The manuscript is about fin fold regeneration in zebrafish embryos. The authors performed scRNAseq analysis of embryonic tails at 24 hours post-amputation (4 days post-fertilization) and control uninjured tails at the corresponding developmental time point. For this analysis, they used triple transgenic fish, which were previously established by other groups. Then, a population of *foxd3:EGFP*-expressing cells is characterized during fin fold regeneration. Functional studies using *foxd3*-morpholinos and previously established *foxd3*-mutants suggested a regeneration phenotype. Subsequently, the recruitment of macrophages is characterized in relation to *foxd3*-positive cells. Finally, pharmacological treatment with PD168393 or AG1478 was used to investigate the *foxd3*+cell-dependent NRG1/ErbB pathway for regeneration in zebrafish embryos.

Overall, the concept of the dialogue between macrophages and neural crest-derived cells during zebrafish fin fold regeneration is intriguing. Enthusiasm was weakened, however, by the lack of new transgenic tools, which would be designed to more precisely address this hypothesis. Furthermore, analyses of the scRNAseq data are not yet presented in a comprehensive manner in order to increase our understanding about the relevant biological processes. The central claims of this study are not always convincingly supported by experimental evidence. The strength of the work is a rich amount of interesting experiments and very nice movies.

Major concerns

1.
scRNAseq data analysis

The analyses of scRNAseq data are incomplete and they are presented in a preliminary form. The following points need to be addressed:

1.1. Quality controls for scRNAseq are missing.

1.2. The authors chose to use triple transgenic zebrafish for scRNAseq, as shown in Fig. 1a: *foxd3:EGFP/rcn3:Gal4/UAS:mCherry*. This approach should result in a molecular characterization of three cell populations that express the relevant transgene (EGFP, Gal4, mCherry). Very surprisingly, scRNAseq analyses do not show any cells identified by EGFP or Gal4/mCherry transcripts ! What has happened?

The authors did not use their scRNAseq data to determine what is the molecular identity of neural crest derived cells (NCdC) that express *foxd3*+:eGFP in the fin, which is the subject of the paper. Do these cells belong to a pigment-cell lineage or Schwann cells, as it was shown in some previous papers?

What about the cells expressing Gal4 and mCherry in this scRNAseq analysis? To which cell types can they be assigned? Do they modulate their gene expression between uninjured and regeneration conditions?

1.3. The authors show two UMAP clusterings: the 1st with integrated data of uncut versus regenerating fins (Fig. 1) and the 2nd with only uncut samples in Suppl. Fig. 1 (I assume that uncut fins were at 4 dpf and not at 3 dpf. The age of control fins is not clearly written).

As shown on the schematic drawing of Fig. 1a, the dissected body part does not only include the caudal fin fold, but should also comprise other tissues of the embryonic tail, such as the posterior notochord, spinal cord, nerves, sensory cells, blood vessels, blood cells, pigment cells. In the 1st UMAP clustering, the authors identified 7 cell populations with differentially expressed genes as shown in Fig. 1.

Surprisingly, they also identified similar 7 cell clusters in control uncut fins (Suppl. Fig. 1), which is difficult to understand:

Page 7: "Finally, in the intact caudal fin fold, the unsupervised UMAP analysis also identified the same seven cell clusters (Supplementary Fig. 1): epidermis (cluster 1), epidermis/mesenchymal cells (cluster 2), apical ectodermal ridge (cluster 3), mesenchymal cells (cluster 4), proliferative cells (cluster 5), SC/NCdC (cluster 6) and myeloid cells (cluster 7).

I do not understand these findings: Where are all the other cell types of the embryonic tails? The authors claim that their methodology was unbiased.

How should we interpret these results?

1.4. It is not clear whether the numbers or the color code assigned to the clusters remain the same between these two clusterings in Fig 1 and Suppl. Fig. 1. The analyses require a better graphical representation with clear cell type labelling on the hierarchical clustering, next to the UMAPs. The current presentation of the data is confusing and it is unclear how to interpret the data.

1.5. Bioinformatic integration of UMAP clusterings should be improved for making comparisons between uninjured and regenerating fin.

1.6. It is essential to provide a well-structured list of all the markers used for clustering different cell types. Appropriate references for these markers must be cited.

1.7. It is necessary to add a list of DE genes for the most relevant subsets of cells, such as those expressing GFP, Gal4/mCherry, foxd3, and for mesenchymal cells and macrophages.

1.8. At the bottom of page 7, there is a statement that "in both amputated and uninjured samples, the proliferative cell population was closer to mesenchymal cell". This statement is unclear and one needs more precisions about this proximity.

1.9. Page 9 "This is in accordance, with the scRNA-seq results that showed a reprogramming in this population, becoming more mesenchymal-like (Fig. 1, Supplementary Fig. 1)."

I do not see evidence for this statement in the indicated figures. The author should explain this statement and make visually understandable on the pointed figures.

2.

Quantification of data during development and regeneration.

This comment refers to most of the figures starting from Fig. 2.

The authors evaluate expression of several transgenes, such as foxd3:eGFP, rcn3:Gal4/UAS:mCherry, mpeg:mCherry and tnfa:eGFP, in many experiments. The regenerative process occurs concomitantly to rapid development between 3 to 6 dpf.

To clearly demonstrate that the observed differences at specific time points are related to the regeneration process and not to ongoing development, it is necessary to know which experimental groups are compared to which controls.

Given that the amputation was performed at 72 hpf, a regeneration time-point of 6 hpa corresponds to uninjured control at 78 hpf; 24 hpa should be compared to uncut fins at 96 hpf, 48 hpa should be

compared to uncut fins at 5 dpf, and 72 hpa to uncut fins at 6 dpf. In the manuscript, it is often not clear if such comparisons have indeed been performed for each experiment presented in this study.

Here I listed several (not all) examples of experiments, in which controls should be revised according to the corresponding developmental time points of uninjured fins.

Fig. 2e, live imaging of the fin fold

Fig. 3a the count of pH3+ cells

Fig. 4b. the contact duration of foxd3+NCdC cells and macrophages

Fig. 5i, j. the count of mpeg+ and mpeg+ tnfa+ cells

Fig. 6b,c , d, e, f, g and Supp. Fig. 6. the expression levels of mCherry+ GFP+

Fig. 2g, h, i. the frequency of mCherry+, GFP+ and colocalizing cells

In addition, I have another comment about figure Fig. 2g-i: The author should specify what is the "frequency unit" shown in y-axis of Fig. 2g, 2h, 2i. Is there any reason why SEM was not shown for uncut samples?

3.

Increase of foxd3:eGFP+ rcn3+ cells during regeneration

3.1. The authors claim that there is an increase of foxd3:eGFP+ rcn3+ cells at 6 hpa and 24 hpa compared to uncut control. This change does not appear clearly on the figure 2e and 2f nor Supp. Fig. 2a. This statement would require better evidence and image quantification at the corresponding developmental time points after and without cutting the fin, as explained in the previous comment.

3.2. Based on the shape of the foxd3:eGFP cells, I am just wondering if it is possible that foxd3:eGFP-cells belong to a lineage of pigment cells?

4.

Change of cell morphology

In Supplementary Fig. S2, the authors quantified a morphology change: increased roundness and circularity of foxd3+rcn3+ cells. It would be necessary to explain the relevance of such changes, as well as the method used to assess it.

5.

Quantification of cell proliferation using phospho-Histone H3 antibody in Fig. 3

The method of calculation shown in Fig. 3a, d, g is not acceptable. A fold change of cut versus uncut is not informative or even misleading in the figure 3a. For example, one could interpret that cell proliferation drops down at 48 hpa. This quantification has to be shown in a form of real numbers of pH3-positive cells in the fin area and, most importantly, in mCherry-expressing cells. Importantly, this quantification should be supported by representative images.

Furthermore, the statement that "foxd3+ cells did not proliferate" should not solely be based on the pH3 marker, which demarcates only a very short phase during the mitosis. The likelihood of catching a cell at this moment of chromosomal segregation is not sufficiently high to make this conclusion that "foxd3+ cells did not proliferate". To support this statement the authors should apply additional classical cell proliferation assays that robustly detect cells in the G1/S phase of the cell cycle. Representative images and quantification should be then provided.

The next sentence also lacks sufficient evidence: "Moreover, all proliferating rcn3+ cells were physically close to foxd3+ NCdC, as revealed by 4D confocal microscopy at 6 hpA in Tg(foxd3:eGFP;rcn3:Gal4/UAS:mCherry) larvae (Fig. 3b)."

Fig. 3b does not show any proliferation marker.

Representative images of immunoassayed cells with proliferation markers and quantification of proliferating cells should be provided to support this statement.

6

The loss of regenerative capacity in *foxd3* morphants and mutants

The graphs in Fig. 3e, f show that "growth of the fin" in control fins was approximately 160 micrometers, whereas in *foxd3*-morphants and *foxd3*-mutants approx. 130 micrometers. The authors interpret this data as "loss of regenerative capacity". To me, this 30 micrometers difference suggests a mild impairment of regeneration or a mild growth delay. We do not know, but maybe the difference of 30 micrometers will be compensated during the subsequent day of regeneration? The conclusion that these data show "a loss of regenerative capacity" sounds like an overstatement, which should be toned down.

7.

foxd3:eGFP and macrophages

7.1. The significance of the interactions between *foxd3*:GFP cells and macrophages is still unclear. Do *foxd3*:GFP-positive cells also increase recruitment of macrophages in intact fins? Is the interaction between *foxd3*⁺ cells and macrophages wound-dependent or wound-independent? Does this interaction occur during normal development at the indicated time points?

7.2. How can we be sure that *foxd3*⁺ cells induce macrophage recruitment and it is not the other way around? Is it possible that immune cells act to increase the recruitment of *foxd3*⁺ cells after injury?

8. Pharmacological treatments with PD168393 and AG1478

The link between the phenotype observed after pharmacological treatments and the function of *foxd3*⁺ cells is unclear. Both drugs block signaling in all tissues of the whole embryonic body, including mesenchymal cells, epidermal cells and immune cells.

8.1. The first issue is the effect of both drugs on development and normal growth, irrespectively of *foxd3*⁺-cells. Given that the drugs globally block the cell proliferation and migration programs in the embryo, it is expected that the fin regeneration as well as fish development are inhibited, even in a *foxd3*-independent manner.

How can we distinguish between global and *foxd3*-dependent effects in these experiments?

The conclusion that "*foxd3*-dependent NRG1/ErbB2 signalling has an essential role in appendage regeneration" requires better evidence.

Minor comments

1.

Supplementary Fig. S3c: The image on the bottom right (showing the head of *MOfoxd3 tg(Foxd3:mCherry)ct110ht*) does not allow a clear comparison with the *MOctrl*, as the yolk sac is too much centered and the head not visible.

2.

A similar sentence occurs on two pages:

Page 11 "Particularly, it has been suggested that SC promote the recruitment of pro-inflammatory macrophages and induce their polarization toward an anti-inflammatory phenotype".

Page 5 "Interestingly, after nerve injury, SC release factors that will promote pro-inflammatory macrophage recruitment and macrophage polarization towards an anti-inflammatory phenotype."

3.

Pages 13-14: The authors used PD168393 and Tyrphostin (AG1478), described as two inhibitors of NRG1/ErbB pathway. More information should be provided about the biochemical and physiological actions of these drugs to understand the inhibitory mechanism.

4. The Tg(foxd3:mCherry)ct110 heterozygote line was used for this study, but the authors did not explain the nature of this mutation. Without more information, it is difficult to interpret the experiments.

5.

Images of fins at 0 hpa (taken immediately after amputation) should be included in this study.

6.

Discussion could be shortened and more focused.

Response to REVIEWER COMMENTS

Referee #1:

The study by Beryl et al., demonstrates that a population of neural crest-derived cells exist in the regenerating zebrafish caudal fin fold that are associated with macrophage recruitment, blastema cell proliferation, and NRG1/ErbB signaling. There is a growing number of examples in which neural crest-derived Schwann cells are mediators of regeneration including salamander limb regeneration and mouse digit regeneration. This paper adds to these compelling lines of evidence with yet another example of the importance of the neural crest during appendage regeneration. There have also been several studies demonstrating the importance of macrophages during appendage regeneration in salamanders, mice, and zebrafish, but the mechanism of their recruitment and exact role is still unclear. Lastly, NRG1 signaling has become a signaling pathway shown to be important for appendage and heart regeneration in multiple species. Again, the mechanism of action for NRG1/ErbB signaling is still a work in progress.

The study by Beryl et al., link all these separate lines of study together in the zebrafish dorsal fin fold model of appendage regeneration. They show that neural crest-derived cells recruited the proper macrophage phenotype necessary for regeneration and do so using the NRG1/ErbB2 signaling. Although the study does not answer all open questions in these links, it is an important step forward for explaining that these three apparently unlinked phenomena are linked with one another in order to coordinate appendage regeneration in mice, salamanders, and zebrafish.

I find the data compelling and the main conclusions supported by the experiments, but I do have some concerns with the particular cells that are being analyzed by the scRNAseq as well as the actual identity of the foxd3+ cells. Please find my comments and concerns below moving from beginning to end of the manuscript:

1) The reference numbering in the following sentence needs to be corrected: “transient accumulation of pro-inflammatory macrophages 178.”

2) Figure 1a shows transgenic animal lines including a foxd3:eGFP reporter and an rcn3 mcherry reporter line, but these are not described in the first Results section. It is confusing to the reader because the presentation of the transgenic lines suggest that some sort of cell selection was performed, but this seems not to be the case in the written text. I am aware of the experimental design, but if the reader moves through the paper in a linear manner, they may be misled with Figure 1a if there is no description in the manuscript text. Furthermore, I know this isn't the case, but as it is presented in Figure 1a, it looks like foxd3 was already known to be a marker of an important cell type. I suggest removing the representation of these transgenic lines in figure 1 of the scRNAseq experimental design.

3) Concerning the following sentence: “We then assigned an identity to the cell clusters that composed the blastema of the regenerating fin fold on the basis of the key marker genes that were significantly enriched in each cluster.” How do the authors know that the seven cell populations observed in the 24 hours sample are found in the blastema rather than uninjured tissues? When comparing Figure 1b with Supplemental Figure 1, they seem to have the same

seven cell populations. Does this mean all seven populations are found in uninjured and blastema tissue with similar expression profiles? Or does it mean non-regenerating tissue was collected in the regenerating sample? I feel some in situ hybridization or IHC would help clear up this distinction, which is an important one when trying to understand appendage regeneration.

4) Based upon the following sentence, it is assumed that *hmx3a* and *irg11* are up-regulated during regeneration. I don't see evidence of this presented in this portion of the results. Furthermore, are similar cells found in the uninjured fin? "Cells in this cluster also strongly expressed *hmx3a* (a gene encoding a protein predicted to have sequence-specific DNA binding activity) and *irg11* (a gene encoding a protein involved in the inflammatory response to wounding), suggesting that this cell population might specifically respond to amputation (Fig 1d-e)."

5) In Figure 1, please include which cell types are represented in each cluster either in Fig. 1B, C, or D. It will be informative to understand the relative contribution of each cell type to the total population.

6) It is unclear why clusters are determined "NCdC/mesenchymal cells and epidermis/mesenchymal cells". Is this because the clusters express markers of each cell type? This should be explained in the results where these cells are first described in a bit more detail.

7) I am surprised no melanophore cell signatures were found in the scRNAseq dataset. Are there no melanophores in the tail fin at this point of development?

8) Did the FACS data presented in Supplemental Figure 2a support the conclusion of increased mcherry/GFP+ cells at 24hpa presented in Figure 2i? I also think the error bars are missing in the uninjured control.

9) I think the authors meant to say "beneath" rather than "beyond" in the following sentence: "Caudal fin fold regeneration involves the critical step of blastemal cell proliferation in the area beyond the apical epithelial cap (AEC) from 6 hpA"

10) When I complete the Results section, I feel like the main conclusion I am missing is the identification of the NCdC cell type. Will these cells become Schwann cells in the adult, or do they generate pigment cells, or even mesenchyme? What is the normal function of these cells? It is most likely not to support regeneration.

Response to Referee #1 comments (in blue):

1) The reference numbering in the following sentence needs to be corrected: "transient accumulation of pro-inflammatory macrophages 178."

Response 1) We fully agree with the reviewer's comment. We modified the reference numbering accordingly and highlighted the modification in yellow (line 89).

2) Figure 1a shows transgenic animal lines including a *foxd3:eGFP* reporter and an *rcn3:mcherry* reporter line, but these are not described in the first Results section. It is confusing to the reader because the presentation of the transgenic lines suggest that some sort of cell selection was performed, but this seems not to be the case in the written text. I am aware of the experimental design, but if the reader moves through the paper in a linear manner, they may be misled with Figure 1a if there is no description in the manuscript text. Furthermore, I know this isn't the case, but as it is presented in Figure 1a, it looks like *foxd3* was already known to be a marker of an important cell type. I suggest removing the representation of these transgenic lines in figure 1 of the scRNAseq experimental design.

Response 2) We agree with the reviewer's comment and we thank him/her for raising this error in Figure 1a. Indeed, we did not use the *foxd3:eGFP* and *rcn3:mCherry* reporter lines but rather wild-type zebrafish (AB) as indicated in the main manuscript. Moreover, as underlined by the reviewer, we did not perform any cell selection as indicated in the written text. Modifications to the experimental design have been performed accordingly.

3) Concerning the following sentence: "We then assigned an identity to the cell clusters that composed the blastema of the regenerating fin fold on the basis of the key marker genes that were significantly enriched in each cluster." How do the authors know that the seven cell populations observed in the 24 hours sample are found in the blastema rather than uninjured tissues? When comparing Figure 1b with Supplemental Figure 1, they seem to have the same seven cell populations. Does this mean all seven populations are found in uninjured and blastema tissue with similar expression profiles? Or does it mean non-regenerating tissue was collected in the regenerating sample? I feel some in situ hybridization or IHC would help clear up this distinction, which is an important one when trying to understand appendage regeneration.

Response 3) We agree with the reviewer and we clarified the sentence: "We then assigned an identity to the cell clusters that composed the blastema of the regenerating fin fold on the basis of the key marker genes that were significantly enriched in each cluster." Indeed, the cell clusters that compose the blastema and the intact fin fold are similar because in the two cases (uncut versus cut) we are in the same tissue (one developing and the other regenerating).

However, while we found the same cell subsets in the two conditions, their molecular signatures were slightly different and the expression profiles of some genes expressed by the different cell subsets differed between conditions. In other words, the cell clusters are the same in the compared tissues (uncut versus cut) but they exhibit different transcriptional profiles. We thus, modified appropriately the quoted sentence. All changes made to the text are highlighted in yellow in the marked copy of revised manuscript (lines 126-128 and 150-166; Supplementary Tables 1, 2, 3 and 4).

Of note, we never collected some non-regenerating tissue in the regenerating sample because the regenerating fin fold is always harvested by collecting the tissue from the amputation plane (initial amputation position) to the new distal fin fold edge.

Regarding the seven cell populations, both "cut" and "uncut" data sets were analyzed independently using the same parameters and arguments. Signature marker genes for each

specific cluster were identified with Cell Ranger against all remaining cells on the basis of a traditional k-means clustering approach across a range of K values, where K is the pre-set number of clusters. A major caveat of this approach in scRNA-seq data analysis is the need for a predetermined number of clusters, which is unknown in many scRNA-seq datasets. Initially, we chose to use the first ten clusters and saw that the number of cells was very low from K-mean = 7. Similarly, to the K-means-based algorithms, the use of the graph-based approach also showed seven clusters to drive our cell-cluster identification. To annotate clusters, marker genes were identified using differential expression (DE) analysis. In this particular case, a 'one-versus-all' design was used to identify each cluster-specific marker genes. DE testing was done between groups: i) the cells in the cluster of interest and ii) all the other cells in the dataset. It identified genes that were upregulated in the cluster of interest compared to the rest of the cells. The top 100 significant markers with the largest average log fold change were retained as the signature for each cluster (Supplementary Table 1 and 4). Finally, well-structured lists of markers were used to assign a cell identity to each cluster. The robustness of these markers was assessed by reviewing the literature and fold change value. These lists are presented in Supplementary Table 2 (lines 129-131).

4) Based upon the following sentence, it is assumed that hmx3a and irg11 are up-regulated during regeneration. I don't see evidence of this presented in this portion of the results. Furthermore, are similar cells found in the uninjured fin? "Cells in this cluster also strongly expressed hmx3a (a gene encoding a protein predicted to have sequence-specific DNA binding activity) and irg11 (a gene encoding a protein involved in the inflammatory response to wounding), suggesting that this cell population might specifically respond to amputation (Fig 1d-e)."

Response 4) We fully agree with the reviewer's comment and reformulated the quoted sentence. We replaced "...strongly expressed..." by "...upregulated..." (line 135). Indeed, these two genes are expressed in the same NCC cluster in the uninjured fin. We performed differential gene expression analysis between cluster 7 (that contains hmx3a or irg11) and the other clusters in cut and uncut samples. The comparison of Log2 fold changes showed an upregulation in favour of the Cut fin samples.

	FeatureID	Log2 Fold Change (Cluster 7 Vs other clusters)	P-Value
hmx3a	Cut	9.82	2.57E-12
	Uncut	7.10	1.61E-09

	FeatureID	Log2 Fold Change (Cluster 7 Vs other clusters)	P-Value
irg11	Cut	8.04	3.46E-11
	Uncut	7.04	2.19E-09

5) In Figure 1, please include which cell types are represented in each cluster either in Fig. 1B, C, or D. It will be informative to understand the relative contribution of each cell type to the total population.

Response 5) As requested by the reviewer, we included all cell types represented in each cluster in the new version of Figure 1. Regarding the relative contribution of each cell type to the total population, please see the response above.

6) It is unclear why clusters are determined “NCdC/mesenchymal cells and epidermis/mesenchymal cells”. Is this because the clusters express markers of each cell type? This should be explained in the results where these cells are first described in a bit more detail.

Response 6) As mentioned above:

To annotate clusters, marker genes were identified using differential expression (DE) analysis. In this particular case, a ‘one-versus-all’ design was used to identify each cluster-specific marker genes. DE testing was done between these groups: i) cells in the cluster of interest and ii) all the other cells in the dataset. It identified genes that were upregulated in the cluster of interest compared to the rest of the cells. The top 100 significant markers with largest average log fold change were retained as the signature for each cluster. Finally, well-structured lists of markers were used to identify each cluster profile (Supplementary Table 2) (lines 129-131).

The robustness of these markers was assessed by reviewing the literature and fold change value.

7) I am surprised no melanophore cell signatures were found in the scRNAseq dataset. Are there no melanophores in the tail fin at this point of development?

Response 7) Indeed, there are some melanophores present in the tail. However, there are mainly around the notochord, and very few are present within the mesenchyme. In the context of our experiments, we have been as precise and reproducible as possible, by collecting a minimum amount of the notochord area and more tissue of the mesenchyme area. This explains why we did not find any melanophore signature in our dataset.

8) Did the FACS data presented in Supplemental Figure 2a support the conclusion of increased mcherry/GFP+ cells at 24hpa presented in Figure 2i? I also think the error bars are missing in the uninjured control.

Response 8) We have completed Supplemental Figure 2 that supports now the conclusion presented in Figure 2i.

Regarding the error bars, they are not present in the figure because it is a fold cut/uncut. Therefore, all results related to the uncut condition are equal to 1.

9) I think the authors meant to say “beneath” rather than “beyond” in the following sentence: “Caudal fin fold regeneration involves the critical step of blastemal cell proliferation in the area beyond the apical epithelial cap (AEC) from 6 hpA”

Response 9) We agree with the reviewer's comment and made appropriate modifications (line 237).

10) When I complete the Results section, I feel like the main conclusion I am missing is the identification of the NCdC cell type. Will these cells become Schwann cells in the adult, or do they generate pigment cells, or even mesenchyme? What is the normal function of these cells? It is most likely not to support regeneration.

Response 10) We agree with the reviewer's comment and propose a conclusion about the function and the fate of these cells. All changes made to the text are highlighted in yellow in the marked copy of the revised manuscript (lines 470-473 and 479-486).

Referee #2 (a technical reviewer commenting on scRNAseq data)

In the manuscript, the authors identified a subset of blastemal cells with Foxd3 positives which regulate macrophage recruitment and polarization via the NRG1/ErbB pathway. This is an interesting study and the authors provided solid evidence to support their finds. It is really a nice work.

Major comments:

(1) This study identified 7 clusters (cell subpopulations, Figure 1B and Supplementary Figure 1A). Are the M1/M2 macrophages part of these 7 clusters or the scRNA-seq data do not have such resolution to investigate macrophages?

(2) The only limitation of this study is that the fin regeneration/major conclusion had a context of early development (3 days post fertilization (dpf)). The fin regeneration process and mechanism can be different for adults (e.g., timing, gene expression patterns).

(3) Unsupervised UMAP analysis identified the same 7 cell clusters in both intact and amputated fins (Figure 1B and Supplementary Figure 1A). However, the cluster identify was based on several marker genes. It is unclear whether these cell clusters are really identical in both intact and amputated fins. The main problem of marker gene based methods to identify cell population (and align cell populations in different conditions) is that the definition of marker genes are context dependent. There are many statistical methods which can compare cell subpopulations in different conditions. For example, the Seurat R package (<https://satijalab.org/seurat/v3.2/integration.html>) can perform a canonical correspondence analysis (CCA) to compare and visualize different subpopulations in different conditions. This can be helpful to compare these 7 cell clusters in intact and amputated fins.

Response to Referee #2 comments in blue:

Major comments:

(1) This study identified 7 clusters (cell subpopulations, Figure 1B and Supplementary Figure 1A). Are the M1/M2 macrophages part of these 7 clusters or the scRNA-seq data do not have such resolution to investigate macrophages?

Unfortunately, we are limited by the resolution and the number of cells to distinguish macrophage subpopulations. Moreover, there is a systematic bias against some type of cells during the sample process, i.e. from the tissue dissociation to the incorporation of droplets in the microfluidic device. This bias could explain why we were not able to recover some other cell subsets present in the blastema, leading us to identify mainly seven clusters. Increasing the cell number for scRNA-sequencing or performing scRNA-sequencing on sorted macrophages would have helped to distinguish macrophage subsets.

(2) The only limitation of this study is that the fin regeneration/major conclusion had a context of early development (3 days post fertilization (dpf)). The fin regeneration process and mechanism can be different for adults (e.g., timing, gene expression patterns).

We agree with the reviewer's comment underlying that the use of zebrafish at 3 dpf leads us to introduce a systematic bias. However, this is a parameter we always take into account, when possible, to be as accurate as possible, for example, by normalizing the data we generate on amputated larvae with intact larvae at exactly the same developmental stage.

(3) Unsupervised UMAP analysis identified the same 7 cell clusters in both intact and amputated fins (Figure 1B and Supplementary Figure 1A). However, the cluster identity was based on several marker genes. It is unclear whether these cell clusters are really identical in both intact and amputated fins. The main problem of marker gene based methods to identify cell population (and align cell populations in different conditions) is that the definition of marker genes are context dependent. There are many statistical methods which can compare cell subpopulations in different conditions. For example, the Seurat R package (<https://satijalab.org/seurat/v3.2/integration.html>) can perform a canonical correspondence analysis (CCA) to compare and visualize different subpopulations in different conditions. This can be helpful to compare these 7 cell clusters in intact and amputated fins.

We thank the reviewer for this relevant comment and for his/her advice. To compare and visualize different subpopulations in different conditions, we used the Cell Ranger software that aggregates outputs from multiple runs of Cell Ranger count, normalizing the runs to the same sequencing depth and then recomputing the feature-barcode matrices and analysis on the combined data. Here, we combined data from "cut" and "uncut" into an experiment-wide feature-barcode matrix and analysis. The Cell Ranger aggr command takes a CSV file specifying a list of Cell Ranger count output files and produces a single feature-barcode matrix containing all data. Then, we identified the key differentially expressed markers for every cluster as having a high ratio of expression among cells within a cluster relative to all other cells in the "cut" samples. Furthermore, using RNA expression patterns found in the public database ZFIN (<https://zfin.org/>), we annotated the most likely cell type for each cluster. Additionally, the robustness of those markers was assessed by reviewing the literature. This approach helped us to compare the seven cell clusters in intact and amputated fins (We modified Supplementary figure S1 in the revised version). All changes made to the text are highlighted in yellow in the marked copy of revised manuscript (lines 147-166).

We have harmonized the colour codes to obtain a better graphical representation with clear cell type labelling.

Reviewer #4 (Remarks to the Author):

The manuscript is about fin fold regeneration in zebrafish embryos. The authors performed scRNAseq analysis of embryonic tails at 24 hours post-amputation (4 days post-fertilization) and control uninjured tails at the corresponding developmental time point. For this analysis, they used triple transgenic fish, which were previously established by other groups. Then, a population of *foxd3:EGFP*-expressing cells is characterized during fin fold regeneration. Functional studies using *foxd3*-morpholinos and previously established *foxd3*-mutants suggested a regeneration phenotype. Subsequently, the recruitment of macrophages is characterized in relation to *foxd3*-positive cells. Finally, pharmacological treatment with PD168393 or AG1478 was used to investigate the *foxd3*+cell-dependent NRG1/ErbB pathway for regeneration in zebrafish embryos.

Overall, the concept of the dialogue between macrophages and neural crest-derived cells during zebrafish fin fold regeneration is intriguing. Enthusiasm was weakened, however, by the lack of new transgenic tools, which would be designed to more precisely address this hypothesis. Furthermore, analyses of the scRNAseq data are not yet presented in a comprehensive manner in order to increase our understanding about the relevant biological processes. The central claims of this study are not always convincingly supported by experimental evidence. The strength of the work is a rich amount of interesting experiments and very nice movies.

Major concerns

1. scRNAseq data analysis

The analyses of scRNAseq data are incomplete and they are presented in a preliminary form. The following points need to be addressed:

1.1. Quality controls for scRNAseq are missing.

1.2. The authors chose to use triple transgenic zebrafish for scRNAseq, as shown in Fig. 1a: *foxd3:EGFP/rcn3:Gal4/UAS:mCherry*. This approach should result in a molecular characterization of three cell populations that express the relevant transgene (EGFP, Gal4, mCherry). Very surprisingly, scRNAseq analyses do not show any cells identified by EGFP or Gal4/mCherry transcripts !
What has happened?

The authors did not use their scRNAseq data to determine what is the molecular identity of neural crest derived cells (NCdC) that express *foxd3*+eGFP in the fin, which is the subject of the paper. Do these cells belong to a pigment-cell lineage or Schwann cells, as it was shown in some previous papers?

What about the cells expressing Gal4 and mCherry in this scRNAseq analysis? To which cell types can they be assigned? Do they modulate their gene expression between uninjured and regeneration conditions?

1.3. The authors show two UMAP clusterings: the 1st with integrated data of uncut versus regenerating fins (Fig. 1) and the 2nd with only uncut samples in Suppl. Fig. 1 (I assume that uncut fins were at 4 dpf and not at 3 dpf. The age of control fins is not clearly written).

As shown on the schematic drawing of Fig. 1a, the dissected body part does not only include the caudal fin fold, but should also comprise other tissues of the embryonic tail, such as the posterior notochord, spinal cord, nerves, sensory cells, blood vessels, blood cells, pigment cells. In the 1st UMAP clustering, the authors identified 7 cell populations with differentially expressed genes as shown in Fig. 1.

Surprisingly, they also identified similar 7 cell clusters in control uncut fins (Suppl. Fig. 1), which is difficult to understand:

Page 7: “Finally, in the intact caudal fin fold, the unsupervised UMAP analysis also identified the same seven cell clusters (Supplementary Fig. 1): epidermis (cluster 1), epidermis/mesenchymal cells (cluster 2), apical ectodermal ridge (cluster 3), mesenchymal cells (cluster 4), proliferative cells (cluster 5), SC/NCdC (cluster 6) and myeloid cells (cluster 7).

I do not understand these findings: Where are all the other cell types of the embryonic tails? The authors claim that their methodology was unbiased. How should we interpret these results?

1.4. It is not clear whether the numbers or the color code assigned to the clusters remain the same between these two clusterings in Fig 1 and Suppl. Fig. 1. The analyses require a better graphical representation with clear cell type labelling on the hierarchical clustering, next to the UMAPs. The current presentation of the data is confusing and it is unclear how to interpret the data.

1.5. Bioinformatic integration of UMAP clusterings should be improved for making comparisons between uninjured and regenerating fin.

1.6. It is essential to provide a well-structured list of all the markers used for clustering different cell types. Appropriate references for these markers must be cited.

1.7. It is necessary to add a list of DE genes for the most relevant subsets of cells, such as those expressing GFP, Gal4/mCherry, foxd3, and for mesenchymal cells and macrophages.

1.8. At the bottom of page 7, there is a statement that “in both amputated and uninjured samples, the proliferative cell population was closer to mesenchymal cell”. This statement is unclear and one needs more precisions about this proximity.

1.9. Page 9 “This is in accordance, with the scRNA-seq results that showed a reprogramming in this population, becoming more mesenchymal-like (Fig. 1, Supplementary Fig. 1).”

I do not see evidence for this statement in the indicated figures. The author should explain this statement and make visually understandable on the pointed figures.

2.

Quantification of data during development and regeneration.

This comment refers to most of the figures starting from Fig. 2.

The authors evaluate expression of several transgenes, such as foxd3:eGFP, rcn3:Gal4/UAS:mCherry, mpeg:mCherry and tnfa:eGFP, in many experiments. The

regenerative process occurs concomitantly to rapid development between 3 to 6 dpf. To clearly demonstrate that the observed differences at specific time points are related to the regeneration process and not to ongoing development, it is necessary to know which experimental groups are compared to which controls.

Given that the amputation was performed at 72 hpf, a regeneration time-point of 6 hpa corresponds to uninjured control at 78 hpf; 24 hpa should be compared to uncut fins at 96 hpf, 48 hpa should be compared to uncut fins at 5 dpf, and 72 hpa to uncut fins at 6 dpf. In the manuscript, it is often not clear if such comparisons have indeed been performed for each experiment presented in this study.

Here I listed several (not all) examples of experiments, in which controls should be revised according to the corresponding developmental time points of uninjured fins.

Fig. 2e, live imaging of the fin fold

Fig. 3a the count of pH3+ cells

Fig. 4b. the contact duration of foxd3+NCdC cells and macrophages

Fig. 5i, j. the count of mpeg+ and mpeg+ tnfa+ cells

Fig. 6b,c , d, e, f, g and Supp. Fig. 6. the expression levels of mCherry+ GFP+

Fig. 2g, h, i. the frequency of mCherry+, GFP+ and colocalizing cells

In addition, I have another comment about figure Fig. 2g-i: The author should specify what is the “frequency unit” shown in y-axis of Fig. 2g, 2h, 2i. Is there any reason why SEM was not shown for uncut samples?

3.

Increase of foxd3:eGFP+ rcn3+ cells during regeneration

3.1. The authors claim that there is an increase of foxd3:eGFP+ rcn3+ cells at 6 hpa and 24 hpa compared to uncut control. This change does not appear clearly on the figure 2e and 2f nor Supp. Fig. 2a. This statement would require better evidence and image quantification at the corresponding developmental time points after and without cutting the fin, as explained in the previous comment.

3.2. Based on the shape of the foxd3:eGFP cells, I am just wondering if it is possible that foxd3:eGFP-cells belong to a lineage of pigment cells?

4.

Change of cell morphology

In Supplementary Fig. S2, the authors quantified a morphology change: increased roundness and circularity of foxd3+rcn3+ cells. It would be necessary to explain the relevance of such changes, as well as the method used to assess it.

5.

Quantification of cell proliferation using phospho-Histone H3 antibody in Fig. 3

The method of calculation shown in Fig. 3a, d, g is not acceptable. A fold change of cut versus uncut is not informative or even misleading in the figure 3a. For example, one could interpret that cell proliferation drops down at 48 hpa. This quantification has to be shown in a form of real numbers of pH3-positive cells in the fin area and, most importantly, in mCherry-expressing cells. Importantly, this quantification should be supported by representative

images.

Furthermore, the statement that “foxd3⁺ cells did not proliferate” should not solely be based on the pH3 marker, which demarcates only a very short phase during the mitosis. The likelihood of catching a cell at this moment of chromosomal segregation is not sufficiently high to make this conclusion that “foxd3⁺ cells did not proliferate”. To support this statement the authors should apply additional classical cell proliferation assays that robustly detect cells in the G1/S phase of the cell cycle.

Representative images and quantification should be then provided.

The next sentence also lacks sufficient evidence: “Moreover, all proliferating rcn3⁺ cells were physically close to foxd3⁺ NCdC, as revealed by 4D confocal microscopy at 6 hpA in Tg(foxd3:eGFP;rcn3:Gal4/UAS:mCherry) larvae (Fig. 3b).”

Fig. 3b does not show any proliferation marker.

Representative images of immunoassayed cells with proliferation markers and quantification of proliferating cells should be provided to support this statement.

6

The loss of regenerative capacity in foxd3 morphants and mutants

The graphs in Fig. 3e, f show that “growth of the fin” in control fins was approximately 160 micrometers, whereas in foxd3-morphants and foxd3-mutants approx. 130 micrometers. The authors interpret this data as “loss of regenerative capacity”. To me, this 30 micrometers difference suggests a mild impairment of regeneration or a mild growth delay. We do not know, but maybe the difference of 30 micrometers will be compensated during the subsequent day of regeneration? The conclusion that these data show “a loss of regenerative capacity” sounds like an overstatement, which should be toned down.

7.

foxd3:eGFP and macrophages

7.1. The significance of the interactions between foxd3:GFP cells and macrophages is still unclear. Do foxd3:GFP-positive cells also increase recruitment of macrophages in intact fins? Is the interaction between foxd3⁺ cells and macrophages wound-dependent or wound-independent? Does this interaction occur during normal development at the indicated time points?

7.2. How can we be sure that foxd3⁺ cells induce macrophage recruitment and it is not the other way around? Is it possible that immune cells act to increase the recruitment of foxd3⁺ cells after injury?

8. Pharmacological treatments with PD168393 and AG1478

The link between the phenotype observed after pharmacological treatments and the function of foxd3⁺ cells is unclear. Both drugs block signaling in all tissues of the whole embryonic body, including mesenchymal cells, epidermal cells and immune cells.

8.1. The first issue is the effect of both drugs on development and normal growth, irrespectively of foxd3⁺-cells. Given that the drugs globally block the cell proliferation and migration programs in the embryo, it is expected that the fin regeneration as well as fish

development are inhibited, even in a *foxd3*-independent manner. How can we distinguish between global and *foxd3*-dependent effects in these experiments? The conclusion that “*foxd3*-dependent NRG1/ErbB2 signalling has an essential role in appendage regeneration” requires better evidence.

Minor comments

1. Supplementary Fig. S3c: The image on the bottom right (showing the head of *MOfoxd3 tg(Foxd3:mCherry)ct110ht*) does not allow a clear comparison with the *MOct1*, as the yolk sac is too much centered and the head not visible.
2. A similar sentence occurs on two pages:
Page 11 “Particularly, it has been suggested that SC promote the recruitment of pro-inflammatory macrophages and induce their polarization toward an anti-inflammatory phenotype”.
Page 5 “Interestingly, after nerve injury, SC release factors that will promote pro-inflammatory macrophage recruitment and macrophage polarization towards an anti-inflammatory phenotype.”
3. Pages 13-14: The authors used PD168393 and Tyrphostin (AG1478), described as two inhibitors of NRG1/ErbB pathway. More information should be provided about the biochemical and physiological actions of these drugs to understand the inhibitory mechanism.
4. The *Tg(foxd3:mCherry)ct110* heterozygote line was used for this study, but the authors did not explain the nature of this mutation. Without more information, it is difficult to interpret the experiments.
5. Images of fins at 0 hpa (taken immediately after amputation) should be included in this study.
6. Discussion could be shortened and more focused.

Response to Referee #4 comments (in blue):

The manuscript is about fin fold regeneration in zebrafish embryos. The authors performed scRNAseq analysis of embryonic tails at 24 hours post-amputation (4 days post-fertilization) and control uninjured tails at the corresponding developmental time point. For this analysis, they used triple transgenic fish, which were previously established by other groups. Then, a population of *foxd3:EGFP*-expressing cells is characterized during fin fold regeneration. Functional studies using *foxd3*-morpholinos and previously established *foxd3*-mutants suggested a regeneration phenotype. Subsequently, the recruitment of macrophages is characterized in relation to *foxd3*-positive cells. Finally, pharmacological treatment with PD168393 or AG1478 was used to investigate the *foxd3*+cell-dependent NRG1/ErbB pathway for regeneration in zebrafish embryos.

Overall, the concept of the dialogue between macrophages and neural crest-derived cells during zebrafish fin fold regeneration is intriguing. Enthusiasm was weakened, however, by the lack of new transgenic tools, which would be designed to more precisely address this hypothesis. Furthermore, analyses of the scRNAseq data are not yet presented in a comprehensive manner in order to increase our understanding about the relevant biological processes. The central claims of this study are not always convincingly supported by experimental evidence. The strength of the work is a rich amount of interesting experiments and very nice movies.

Major concerns

1. scRNAseq data analysis

The analyses of scRNAseq data are incomplete and they are presented in a preliminary form. The following points need to be addressed:

1.1. Quality controls for scRNAseq are missing.

We thank the reviewer for addressing this point. We used the Cell Ranger `mkfastq` and `cellranger count` pipelines from the Cell Ranger Single Cell Software by 10x Genomics (<http://10xgenomics.com>) for the initial quality control, sample demultiplexing, mapping, and quantification of raw sequencing data. Mean reads per cell, median number of genes per cell, number of reads, sequencing saturation, percentage of reads mapped confidently to the transcriptome, and other quality metrics were obtained from Cell Ranger output `metrics.csv` files. The confidently mapped reads to intronic, exonic, and intergenic regions were further studied by extracting the number of reads mapping confidently for every cell barcode. The summary of scRNA-seq quality metrics for “cut” and “uncut” samples have been listed in Supplementary Table 5 in the revised version of the manuscript (lines 541-545 and 954-958).

1.2. The authors chose to use triple transgenic zebrafish for scRNAseq, as shown in Fig. 1a: `foxd3:EGFP/rcn3:Gal4/UAS:mCherry`. This approach should result in a molecular characterization of three cell populations that express the relevant transgene (EGFP, Gal4, mCherry). Very surprisingly, scRNAseq analyses do not show any cells identified by EGFP or Gal4/mCherry transcripts !

What has happened?

We agree with the reviewer's comment and we thank him/her for detecting this error in Figure 1a. As also noticed and corrected by Reviewer 1, we did not use the `Tg(foxd3:EGFP/rcn3:Gal4/UAS:mCherry)`. Indeed, we did not use the `foxd3:eGFP` and `rcn3:mCherry` reporter lines but rather wild-type zebrafish (AB) as indicated in the main manuscript. The corresponding modifications to the experimental design have been introduced accordingly in the novel version of the Figure 1 (Fig. 1a).

The authors did not use their scRNAseq data to determine what is the molecular identity of neural crest derived cells (NCdC) that express `foxd3+:eGFP` in the fin, which is the subject of

the paper. Do these cells belong to a pigment-cell lineage or Schwann cells, as it was shown in some previous papers?

Indeed, there are some melanophores present in the tail. However, there are mainly around the notochord, and very few are present within the mesenchyme. In the context of our experiments, we have been as precise and reproducible as possible, by collecting a minimum amount of the notochord area and more tissue of the mesenchyme area. This explains why we did not find any melanophore signature in our dataset.

What about the cells expressing Gal4 and mCherry in this scRNAseq analysis? To which cell types can they be assigned? Do they modulate their gene expression between uninjured and regeneration conditions?

As indicated above, we did not use the *Tg(foxd3:EGFP/rcn3:Gal4/UAS:mCherry)* as initially indicated (our mistake) in Fig.1a. We performed the corresponding modifications to the experimental design to avoid any confusion.

1.3. The authors show two UMAP clusterings: the 1st with integrated data of uncut versus regenerating fins (Fig. 1) and the 2nd with only uncut samples in Suppl. Fig. 1 (I assume that uncut fins were at 4 dpf and not at 3 dpf. The age of control fins is not clearly written).

In the design of all our experiments, as indicated in the presented workflows, the control (intact) fins are always treated exactly as the amputated fins, i.e., we always collected the control and the experimental tissues at the exact same time. Therefore, when the cut fins are collected at 4dpf, the control fins are also collected at 4dpf.

As shown on the schematic drawing of Fig. 1a, the dissected body part does not only include the caudal fin fold, but should also comprise other tissues of the embryonic tail, such as the posterior notochord, spinal cord, nerves, sensory cells, blood vessels, blood cells, pigment cells. In the 1st UMAP clustering, the authors identified 7 cell populations with differentially expressed genes as shown in Fig. 1.

Surprisingly, they also identified similar 7 cell clusters in control uncut fins (Supp. Fig. 1), which is difficult to understand:

Page 7: “Finally, in the intact caudal fin fold, the unsupervised UMAP analysis also identified the same seven cell clusters (Supplementary Fig. 1): epidermis (cluster 1), epidermis/mesenchymal cells (cluster 2), apical ectodermal ridge (cluster 3), mesenchymal cells (cluster 4), proliferative cells (cluster 5), SC/NCdC (cluster 6) and myeloid cells (cluster 7).

I do not understand these findings: Where are all the other cell types of the embryonic tails?

The authors claim that their methodology was unbiased.

How should we interpret these results?

As we indicated in Figure 1a, we did not dissect the whole tails, but only the fin fold located, after the notochord. This part of the collected fin fold is mainly undifferentiated mesenchyme at this developmental stage (doi.org/10.1371/journal.pone.0051766). Therefore, the tissues collected for our study do not contain notochord, spinal cords, blood cells or vessels. Pigments are found in a more proximal area and are not the main population here, in agreement with what we found.

1.4. It is not clear whether the numbers or the color code assigned to the clusters remain the same between these two clusterings in Fig 1 and Supp. Fig. 1. The analyses require a better graphical representation with clear cell type labelling on the hierarchical clustering, next to the UMAPs. The current presentation of the data is confusing and it is unclear how to interpret the data.

We thank the reviewer for noticing this point. To facilitate data interpretation, Supplementary Figure 1 was modified to harmonize the colour codes with Fig.1 in order to obtain a better graphical representation with clear cell type labelling.

1.5. Bioinformatic integration of UMAP clusterings should be improved for making comparisons between uninjured and regenerating fin.

The UMAP clustering has been improved for making comparisons between uninjured and regenerating fin and Supplemental Figure 2 has been added.

1.6. It is essential to provide a well-structured list of all the markers used for clustering different cell types. Appropriate references for these markers must be cited.

In addition to supplementary lists of all clusters, we agree with the reviewer that the well-structured lists of the markers used for clustering different cell types are necessary. These lists are now provided in Supplementary Table 2 of the revised version.

1.7. It is necessary to add a list of DE genes for the most relevant subsets of cells, such as those expressing GFP, Gal4/mCherry, *foxd3*, and for mesenchymal cells and macrophages.

As indicated above, we did not use the *Tg(foxd3:EGFP/rcn3:Gal4/UAS:mCherry)* line but rather a wild-type line (Reviewer's comment 1.2). However, cluster details are provided in Supplementary Table 1 and 4. Regarding the aggregation between cut and uncut samples we have provided the lists of DE genes (Supplementary Table 3) as requested by the reviewer (lines 152-154).

1.8. At the bottom of page 7, there is a statement that "in both amputated and uninjured samples, the proliferative cell population was closer to mesenchymal cell". This statement is unclear and one needs more precisions about this proximity.

Gene ontology (GO) annotations were used to identify the biological processes and functional properties of the 100 genes identified in the cluster 3 and cluster 5 that included mesenchymal cells and the proliferative cell population respectively. Enriched GO terms revealed that numerous functional annotations implicated in metabolic process, skeletal system development and cytoskeleton organization were shared by cluster 3 and cluster 5. This may explain, in part, why the proliferative cell populations were closer to mesenchymal cell.

Accordingly, we removed the initial statement “in both amputated and uninjured samples, the proliferative cell population was closer to mesenchymal cell”. More results are presented lines 211-215 of the revised version of the manuscript.

1.9. Page 9 “This is in accordance, with the scRNA-seq results that showed a reprogramming in this population, becoming more mesenchymal-like (Fig. 1, Supplementary Fig. 1).”

I do not see evidence for this statement in the indicated figures. The author should explain this statement and make visually understandable on the pointed figures.

We agree with the reviewer’s comment and modified this statement (lines 189-191).

2.

Quantification of data during development and regeneration.

This comment refers to most of the figures starting from Fig. 2.

The authors evaluate expression of several transgenes, such as *foxd3:eGFP*, *rcn3:Gal4/UAS:mCherry*, *mpeg:mCherry* and *tnfa:eGFP*, in many experiments. The regenerative process occurs concomitantly to rapid development between 3 to 6 dpf. To clearly demonstrate that the observed differences at specific time points are related to the regeneration process and not to ongoing development, it is necessary to know which experimental groups are compared to which controls.

Given that the amputation was performed at 72 hpf, a regeneration time-point of 6 hpa corresponds to uninjured control at 78 hpf; 24 hpa should be compared to uncut fins at 96 hpf, 48 hpa should be compared to uncut fins at 5 dpf, and 72 hpa to uncut fins at 6 dpf. In the manuscript, it is often not clear if such comparisons have indeed been performed for each experiment presented in this study.

We agree with the reviewer that in our study it is necessary to know which experimental groups are compared to which controls. For this reason, in the different figures, we added workflows, to show that for each experiment, control fins are always treated exactly as amputated fins, i.e., we always collected the control and the experimental tissues at exactly the same time. Therefore, when the cut fins are collected at 78hpf, the control fins are also collected at 78dpf.

Here I listed several (not all) examples of experiments, in which controls should be revised according to the corresponding developmental time points of uninjured fins.

Fig. 2e, live imaging of the fin fold

We understand the reviewer’s comment. However, we would like to underline that all the controls have been integrated in the data analysis. For Fig. 2e, the objective of the experiment was to follow the same fin during the regeneration time course. Therefore, in that context, we cannot provide the uncut condition at the different time points.

Fig. 3a the count of pH3+ cells

Regarding the count of pH3+ cells it is a change fold as indicated in Fig. 3a. All controls have been fixed at each time points and stained in parallel with the zebrafish that underwent caudal

fin fold amputation. The resulting fold change presented in Fig. 3a provides a clear view of the proliferative events occurring during the regeneration process and eliminate the developmental features. We and others have observed a peak of proliferation at 24hpa during the time course of regeneration (doi: 10.7554/eLife.07288;https://doi.org/10.1038/cddis.2017.374). Therefore, in the present article, we used exactly the same method, validated in the quoted articles, to count pH3+ cells.

Fig. 4b. the contact duration of *foxd3*+NCdC cells and macrophages

We and others have shown that in response to caudal fin fold amputation, macrophages are recruited at the amputation site where they polarize toward a M1-like phenotype characterized by expression of *tnfa*. As previously shown (doi: 10.7554/eLife.07288), macrophages, absent in the intact caudal fin fold, are recruited within the first 30minutes of the caudal fin fold regeneration process. Therefore, we cannot assess the contact duration of this macrophage population and the *foxd3*+ NCdC because in the intact caudal fin fold, macrophages are not actively migrating.

Fig. 5i, j. the count of *mpeg*+ and *mpeg*+ *tnfa*+ cells

As explained above and as previously published (doi: 10.7554/eLife.07288 ;https://doi.org/10.1038/cddis.2017.374), there are no *mpeg*+ (or negligible non migrating cells) and no *mpeg*+ *tnfa*+ cells in the intact caudal fin fold. Therefore, this tissue cannot serve as a control.

Fig. 6b,c , d, e, f, g and Supp. Fig. 6. the expression levels of *mCherry*+ *GFP*+

As previously published (doi: 10.7554/eLife.07288;https://doi.org/10.1038/cddis.2017.374), in intact caudal fin fold of zebrafish larvae, there is no polarization of macrophages, so we cannot sort *mCherry*+ *GFP*+cells in intact fins.

Fig. 2g, h, i. the frequency of *mCherry*+, *GFP*+ and colocalizing cells

In addition, I have another comment about figure Fig. 2g-i: The author should specify what is the “frequency unit” shown in y-axis of Fig. 2g, 2h, 2i. Is there any reason why SEM was not shown for uncut samples?

As indicated in the histograms, the y-axis shows a fold change. There is no SEM for the uncut samples because the fold change is normalized to 1.

3.

Increase of *foxd3*:eGFP+ *rcn3*+ cells during regeneration

3.1. The authors claim that there is an increase of *foxd3*:eGFP+ *rcn3*+ cells at 6 hpa and 24 hpa compared to uncut control. This change does not appear clearly on the figure 2e and 2f nor Supp. Fig. 2a. This statement would require better evidence and image quantification at the corresponding developmental time points after and without cutting the fin, as explained in the previous comment.

We agree with the reviewer's comment. Indeed, such a result requires image quantification and of course a comparative analysis of the cut and uncut conditions. We thus performed some image quantification, and we added a FACS analysis to have a more robust quantification. As indicated in the manuscript and the figures, for the quantification we always compared the uncut and cut conditions (referred as the fold cut/uncut).

3.2. Based on the shape of the *foxd3:eGFP* cells, I am just wondering if it is possible that *foxd3:eGFP*-cells belong to a lineage of pigment cells?

Indeed, it is possible that at least some of *foxd3*⁺ cells could belong to the pigment lineage. However, it is worth to consider, as mentioned previously, that the vast majority of pigment cells is localized in a more proximal area than the one used for the experiments, and that many of these cells are within the mesenchyme, exhibiting a similar morphology as *mcherry*⁺ mesenchymal cells.

4.

Change of cell morphology

In Supplementary Fig. S2, the authors quantified a morphology change: increased roundness and circularity of *foxd3*⁺*rcn3*⁺ cells. It would be necessary to explain the relevance of such changes, as well as the method used to assess it.

We agree with the reviewer's comment, and thus propose, in the new version of the manuscript, an explanation to such changes.

Morphological changes have been assessed with the Fiji software using the circularity and roundness plugins. These plugins are an extended version of the ImageJ Measure command that calculates object circularity using the formula: $\text{Circularity} = 4\pi(\text{area}/\text{perimeter}^2)$.

A circularity value of 1.0 indicates a perfect circle. As the value approaches 0.0, it indicates an increasingly elongated polygon.

$\text{Roundness} = 4\text{area}/(\pi\text{major_axis}^2)$.

Thus, roundness is more relative to the area of the object compared to the main axis. It could be considered as opposite to elongation factor, whereas the circularity refers to the shape of the object compared to a perfect circle.

All changes made to the text are highlighted in yellow in the marked copy of revised manuscript (lines 602-609).

5.

Quantification of cell proliferation using phospho-Histone H3 antibody in Fig. 3

The method of calculation shown in Fig. 3a, d, g is not acceptable. A fold change of cut versus uncut is not informative or even misleading in the figure 3a. For example, one could interpret that cell proliferation drops down at 48 hpa. This quantification has to be shown in a form of real numbers of pH3-positive cells in the fin area and, most importantly, in mCherry-

expressing cells. Importantly, this quantification should be supported by representative images.

Indeed, for each set of results, we have the uncut data as control. However, we decided not to present these data in the present article because they are statistically not different. Therefore, as we did not find any difference of the cell proliferation rate in a developmental context (uncut developing fin fold) when we compared the morphants, the mutants and the wild type larvae, we presented the fold change of “cut versus uncut”. This method to analyse the cell proliferation rate in the regenerating caudal fin has been previously validated by us and others (doi: 10.7554/eLife.07288; <https://doi.org/10.1038/cddis.2017.374>; doi: 10.1242/dev.098459; doi: 10.1002/dvdy.20181). In that way, we showed that the well-described peak of proliferation at 24hpA might be affected in response to *foxd3* expression level modification.

Furthermore, the statement that “*foxd3*⁺ cells did not proliferate” should not solely be based on the pH3 marker, which demarcates only a very short phase during the mitosis. The likelihood of catching a cell at this moment of chromosomal segregation is not sufficiently high to make this conclusion that “*foxd3*⁺ cells did not proliferate”. To support this statement the authors should apply additional classical cell proliferation assays that robustly detect cells in the G1/S phase of the cell cycle.

Representative images and quantification should be then provided.

We agree with the reviewer’s comment and thus qualified our statement. To that end, we specified in the new version of our manuscript that *foxd3*⁺ cells did not proliferate upon amputation because the fold change was close to 1, as indicated in Fig. 3a (lines 212-216). A representative image of PH3 staining in a *Tg(foxd3:eGFP)* larva is provided in Fig. S3a.

The next sentence also lacks sufficient evidence: “Moreover, all proliferating *rcn3*⁺ cells were physically close to *foxd3*⁺ NCdC, as revealed by 4D confocal microscopy at 6 hpA in *Tg(foxd3:eGFP;rcn3:Gal4/UAS:mCherry)* larvae (Fig. 3b).”

We agree with the reviewer’s comment. This sentence has been suppressed. More analyses have been performed and the results are presented in the Supplementary Fig. 3 and discussed in the text (lines 212-216).

Fig. 3b does not show any proliferation marker.

Representative images of immunoassayed cells with proliferation markers and quantification of proliferating cells should be provided to support this statement.

We agree with the reviewer’s comment. For this reason, we added Movie 1 to complete Fig. 3b that has been extracted from Movie 1. In this movie, we can clearly see the division of mesenchymal cells during regeneration. Cell proliferation was quantified and added in Fig. 3a.

6

The loss of regenerative capacity in *foxd3* morphants and mutants

The graphs in Fig. 3e, f show that “growth of the fin” in control fins was approximately 160

micrometers, whereas in *foxd3*-morphants and *foxd3*-mutants approx. 130 micrometers. The authors interpret this data as “loss of regenerative capacity”. To me, this 30 micrometers difference suggests a mild impairment of regeneration or a mild growth delay. We do not know, but maybe the difference of 30 micrometers will be compensated during the subsequent day of regeneration? The conclusion that these data show “a loss of regenerative capacity” sounds like an overstatement, which should be toned down.

We fully agree with the reviewer’s comment and we thank him/her. Therefore, we appropriately modified the interpretation of the results presented in Fig. 3e to underline that we did not observe a “loss of regenerative capacity” but only an impairment (lines 248-249).

7.

foxd3:eGFP and macrophages

7.1. The significance of the interactions between *foxd3*:GFP cells and macrophages is still unclear. Do *foxd3*:GFP-positive cells also increase recruitment of macrophages in intact fins? Is the interaction between *foxd3*⁺ cells and macrophages wound-dependent or wound-independent? Does this interaction occur during normal development at the indicated time points?

As indicated above, in the intact caudal fin fold, macrophages are not recruited.

7.2. How can we be sure that *foxd3*⁺ cells induce macrophage recruitment and it is not the other way around? Is it possible that immune cells act to increase the recruitment of *foxd3*⁺ cells after injury?

As *foxd3*⁺ cells are already present in intact fin fold and this number is not increased upon amputation, we can affirm that there is no migration or recruitment of this cell population and that immune cells do not act to increase the recruitment of *foxd3*⁺ cells after injury.

8. Pharmacological treatments with PD168393 and AG1478

The link between the phenotype observed after pharmacological treatments and the function of *foxd3*⁺ cells is unclear. Both drugs block signaling in all tissues of the whole embryonic body, including mesenchymal cells, epidermal cells and immune cells.

8.1. The first issue is the effect of both drugs on development and normal growth, irrespectively of *foxd3*⁺-cells. Given that the drugs globally block the cell proliferation and migration programs in the embryo, it is expected that the fin regeneration as well as fish development are inhibited, even in a *foxd3*-independent manner.

How can we distinguish between global and *foxd3*-dependent effects in these experiments? The conclusion that “*foxd3*-dependent NRG1/ErbB2 signalling has an essential role in appendage regeneration” requires better evidence.

We agree that this point is pivotal in our study. Therefore, we provided evidence for this statement. As indicated in the main manuscript, “to determine which blastema cells expressed

nrg1, we focused on NCdC and performed in situ hybridization in control and foxd3 morphants at 24 hpA. Foxd3 deficiency was associated with massive nrg1 downregulation. Of note, the four zebrafish nrg1 variants were not differentially expressed in response to amputation in *Tg(foxd3:mCherry)^{ct110}* mutant larvae (Fig. 5f-g and data not shown). Moreover, nrg1.004 expression levels were comparable in non-amputated *Tg(foxd3:mCherry)^{ct110}* and wild type larvae. In *Tg(foxd3:mCherry)^{ct110}* mutants, erbb2 and erbb3 expression levels were not different compared with those in WT larvae and in response to amputation (Fig. 5g and Supplementary Fig. 5a). These results suggest that the significant erbb2 and nrg1.004 upregulation observed in wild type larvae upon amputation depends on foxd3, and that foxd3-dependent NRG1/ErbB2 signaling has an essential role in appendage regeneration, by highlighting the correlation between nrg1 expression and presence of foxd3⁺ cells in the regenerating blastema”.

Minor comments

1.

Supplementary Fig. S3c: The image on the bottom right (showing the head of MOfoxd3 tg(Foxd3:mCherry)ct110ht) does not allow a clear comparison with the MOct1, as the yolk sac is too much centered and the head not visible.

We thank the reviewer for his/her suggestion and we made the panel more reader-friendly by clearly indicating the structures observed here (panel e Supplementary Fig. S3).

However, the purpose of this panel is to emphasize the disappearance of mCherry signal in MOfoxd3 injected tg(Foxd3:mCherry)ct110ht embryos. Indeed, this line expresses a mCherry fusion protein with FoxD3, showing how the pattern of the endogenous FoxD3 protein can be modified using a knock-down approach. The expected effect on the head skeleton of the MOfoxd3 is clearly visible on panel d of Supplementary Fig. S3: the tg(col2a:mcherry) allowed us to focus on this particular neural crest-derived structure and on the general phenotype of the morphants.

2.

A similar sentence occurs on two pages:

Page 11 “Particularly, it has been suggested that SC promote the recruitment of pro-inflammatory macrophages and induce their polarization toward an anti-inflammatory phenotype”.

Page 5 “Interestingly, after nerve injury, SC release factors that will promote pro-inflammatory macrophage recruitment and macrophage polarization towards an anti-inflammatory phenotype.”

We agree with the reviewer’s comment and we thank him/her for raising this point. That has been corrected (lines 98-100).

3.

Pages 13-14: The authors used PD168393 and Tyrphostin (AG1478), described as two inhibitors of NRG1/ErbB pathway. More information should be provided about the biochemical and physiological actions of these drugs to understand the inhibitory mechanism. To address the reviewer’s comment, we inserted two references (line 311).

4. The Tg(foxd3:mCherry)ct110 heterozygote line was used for this study, but the authors did not explain the nature of this mutation. Without more information, it is difficult to interpret the experiments.

To address the reviewer's comment, we inserted reference 30 in our manuscript (lines 230).

5.

Images of fins at 0 hpa (taken immediately after amputation) should be included in this study.

An image of fin at 0 hpA has been added in Figure 2e.

6.

Discussion could be shortened and more focused.

We made some modifications in the Discussion. One paragraph has been removed and all changes made to the text are highlighted in yellow in the marked copy of the revised manuscript (lines 470-473 and 479-486).

REVIEWERS' COMMENTS

Reviewer #1 (Remarks to the Author):

The authors have considered all comments I made in my review and made appropriate changes in the manuscript. I have no further comments on the manuscript.

Sincerely,

James Monaghan

Reviewer #4 (Remarks to the Author):

The manuscript has been revised and the all the comments have been carefully addressed. Although the manuscript has been improved and corrected, several major concerns still remain open.

1.

scRNA seq: Comparison between the original and new versions (Fig. 1).

It is disappointing that the triple transgenic fish (as reported in the original Fig. 1a) were in fact wild type fish in scRNA-seq analysis. The authors have responded:

"we did not use the Tg(foxd3:EGFP/rcn3:Gal4/UAS:mCherry) as initially indicated (our mistake) in Fig.1a."

It is a pity because we have missed an opportunity to get more knowledge about the identity and molecular dynamics of foxd3:EGFP and rcn3:Gal4/UAS:mCherry labeled cells, which are the main focus of the study. Thus, the switch from transgenic to wild type fish markedly weakens the potential impact of the scRNA-seq data.

After correcting the mistake with the fish strain, the authors have now also modified another methodological aspect of their scRNAseq experiment, namely the illustration of the dissected tissue. In the rebuttal letter, the authors have answered:

"As we indicated in Figure 1a, we did not dissect the whole tails, but only the fin fold located after the notochord".

Why did the authors originally show a larger amount of tissue in the same figure of the first submission?

Do the other schematic drawings of zebrafish tails also need to be adjusted in the subsequent figures, according to the Fig. 1a?

In the abstract, the authors claim that they "established the first detailed cell atlas of the regenerating caudal fin in zebrafish larvae." Is this detailed cell atlas indeed including all regeneration-relevant cell types throughout the regenerative process?

Is it possible that the source of fin blastema might arise from a region at the level of the notochord, and these cells are not included in the current scRNA-seq analysis?

2

Live imaging of triple transgenic fish in FIG.2e

It is also disappointing that the authors cannot add a control panel to show uninjured triple transgenic fish at the relevant time points, as they responded "we cannot provide the uncut conditions at the different time points".

I understand that "the objective of the experiment was to follow the same fin during the regeneration

time course". Nevertheless, the same time course can be easily applied for control fish.

Given that regeneration is concomitant of development between 3 and 6 dpf, it is necessary to include uninjured control at the corresponding days. Without control, we cannot conclude if the restorative process recapitulates normal developmental dynamics or it involves new regeneration-specific features. To me, this knowledge is essential to understand the regenerative process, when it occurs during rapid development.

Therefore, I believe the readers of this paper would really appreciate to be able to compare images of larval transgenic tails between controls and regeneration.

72 hpf uninjured as control of 0 hpA

78 hpf uninjured as control of 6 hpA

4 dpf uninjured as control of 24 hpA

5 dpf uninjured as control of 48 hpA

6 dpf uninjured as control of 72 hpA

In any way, these developmental time points are used as uninjured controls for subsequent experiments, as confirmed by authors in the response letter. Therefore, it is important to show these fins.

3.

foxd3- morphant phenotype

The size of regenerated fin was about 160 micrometers in control, whereas it was 130 micrometers in foxd3-morphants.

The authors agreed that these data show only a mild impairment of regeneration, but not a "loss of regenerative capacity", as claimed in the previous version. They have now reformulated the conclusion in the results.

Nevertheless, the abstract is still the same as in the first version and the conclusions have not been revised. We still read: "Genetic depletion of these foxd3-positive neural crest-derived cells showed that they are required for blastema formation and caudal fin regeneration."

This statement is not supported by the data of the revised manuscript.

4.

foxd3-positive blastemal cells

In the response letter, the author confirm that they "specified in the new version of our manuscript that fox3+ cells did not proliferate upon amputation" and later "foxd3+ cells are already present in intact fin fold and this number is not increased upon amputation".

However, the abstract still contains a previous conclusion about a discovery of "a novel foxd3-positive blastemal cell population". It is confusing given that the key feature of blastemal cells is their highly proliferative and less differentiated character, required for regeneration.

The claim that foxd3-positive cells can be considered as blastemal cells is not supported by evidence.

In general, I have an impression that several main conclusions/interpretations, especially in the abstract, should be revised according to the results.

Reviewer #5 (Remarks to the Author):

Referee #2 (who reviewed the technical aspects of the scRNA-seq analysis) could not respond to the

author's changes, and so I have been asked to assess whether or not the concerns noted by this reviewer were addressed satisfactorily. I believe that they have.

Notably, the Reviewer was concerned that the marker genes used by the authors are not robust and therefore suggested that the authors use a "cell integration" approach (e.g., the CCA-based approach implemented in Seurat) to annotate their cells. While in general it is true that the marker genes used to classify many cell types are not well-standardized, the authors argue that their cell markers are robust. While I don't have the expertise required to assess whether the specific zebra fish cell type markers in this study are indeed robust and discriminative, I agree with the authors that in general, this is an acceptable way to perform their cell type classification so long as there is evidence that these cell markers are discriminative. The author's inclusion of Supplementary Table 2, which provides references for each cell type marker, is very important for reproducibility and provides readers with a justification for the cell type labels assigned by the authors.

Reviewer #4 (Remarks to the Author):

The manuscript has been revised and the all the comments have been carefully addressed. Although the manuscript has been improved and corrected, several major concerns still remain open.

1.

scRNA seq: Comparison between the original and new versions (Fig. 1).

It is disappointing that the triple transgenic fish (as reported in the original Fig. 1a) were in fact wild type fish in scRNA-seq analysis. The authors have responded:

“we did not use the Tg(*foxd3:EGFP/rcn3:Gal4/UAS:mCherry*) as initially indicated (our mistake) in Fig.1a.”

It is a pity because we have missed an opportunity to get more knowledge about the identity and molecular dynamics of *foxd3:EGFP* and *rcn3:Gal4/UAS:mCherry* labeled cells, which are the main focus of the study. Thus, the switch from transgenic to wild type fish markedly weakens the potential impact of the scRNA-seq data.

The authors understand the reviewer comment. However, the initial aim of the study was to determine the cellular and transcriptomic profiles of intact (uninjured) and regenerating caudal fin fold without a priori consideration for a particular population of cells. It is the reason why the authors have used wild zebrafish instead of a transgenic line. Such an approach has allowed the authors to identify a cell population that corresponds to *foxd3⁺sox10⁺erg2b⁺NCdC*, a cell population that the authors focused on in the rest of the manuscript but not in Figure 1.

After correcting the mistake with the fish strain, the authors have now also modified another methodological aspect of their scRNAseq experiment, namely the illustration of the dissected tissue.

In the rebuttal letter, the authors have answered:

“As we indicated in Figure 1a, we did not dissect the whole tails, but only the fin fold located after the notochord”.

Why did the authors originally show a larger amount of tissue in the same figure of the first submission?

To address the reviewer’s comment (in the first revision of the manuscript) that had arisen from the initial schematic representation of the author’s experimental approach (presented in the first version of the Fig. 1a) and to avoid any confusion, the authors have accordingly improved the workflow in the revised version of the manuscript to be as precise as possible regarding the tissue collected for the study. Under no circumstances the authors modified the method.

The actual Fig. 1a is the most representative of the experimental approach: namely a dissection of a maximum of tissue while avoiding as much as possible the notochord. Of note, due to the small size of the collected tissue a negligible number of notochord cells, as indicated in the transected plan of Fig.1a, appeared in the scRNAseq analysis.

Do the other schematic drawings of zebrafish tails also need to be adjusted in the subsequent figures, according to the Fig. 1a?

The authors thank the reviewer for raising this point. All the schematic drawings of zebrafish tails in the different figures are correct and do not need to be adjusted.

In the abstract, the authors claim that they “established the first detailed cell atlas of the regenerating caudal fin in zebrafish larvae.” Is this detailed cell atlas indeed including all regeneration-relevant cell types throughout the regenerative process?
Is it possible that the source of fin blastema might arise from a region at the level of the notochord, and these cells are not included in the current scRNA-seq analysis?

The authors understand the point raised by the reviewers. However, in all experiments performed in the study, the fin fold has always been amputated below the level of the notochord (i.e., the notochord has always been avoided as much as possible when collecting the samples). The authors have done several movies using transgenic zebrafish lines allowing to track notochord cells and never observed any cells from the notochord migrating to /accumulating within the blastema when this structure was intact. The only cells arising from other regions than the fin fold in the fin blastema were immune cells (neutrophils and macrophages).

2

Live imaging of triple transgenic fish in FIG.2e

It is also disappointing that the authors cannot add a control panel to show uninjured triple transgenic fish at the relevant time points, as they responded “we cannot provide the uncut conditions at the different time points”.

I understand that “the objective of the experiment was to follow the same fin during the regeneration time course”. Nevertheless, the same time course can be easily applied for control fish.

Given that regeneration is concomitant of development between 3 and 6 dpf, it is necessary to include uninjured control at the corresponding days. Without control, we cannot conclude if the restorative process recapitulates normal developmental dynamics or it involves new regeneration-specific features. To me, this knowledge is essential to understand the regenerative process, when it occurs during rapid development.

Therefore, I believe the readers of this paper would really appreciate to be able to compare images of larval transgenic tails between controls and regeneration.

72 hpf uninjured as control of 0 hpA

78 hpf uninjured as control of 6 hpA

4 dpf uninjured as control of 24 hpA

5 dpf uninjured as control of 48 hpA

6 dpf uninjured as control of 72 hpA

In any way, these developmental time points are used as uninjured controls for subsequent experiments, as confirmed by authors in the response letter. Therefore, it is important to show these fins.

The authors agree with the reviewer and show, as requested, in Supplementary Figure 2a the uncut/control fin fold images at the different time points. The corresponding modification made to the text has been introduced accordingly in the revised version of the manuscript (lines 172 and 914-917).

3.

foxd3- morphant phenotype

The size of regenerated fin was about 160 micrometers in control, whereas it was 130 micrometers in *foxd3*-morphants.

The authors agreed that these data show only a mild impairment of regeneration, but not a “loss of regenerative capacity”, as claimed in the previous version. They have now reformulated the conclusion in the results.

Nevertheless, the abstract is still the same as in the first version and the conclusions have not been revised. We still read: “Genetic depletion of these *foxd3*-positive neural crest-derived cells showed that they are required for blastema formation and caudal fin regeneration.”

This statement is not supported by the data of the revised manuscript.

The authors fully agree with the reviewer’s comment and thank her/him for raising this point. They accordingly revised the abstract and replaced the previous sentence by “Genetic depletion of these *foxd3*-positive neural crest-derived cells (NCdC) showed that they are involved for blastema formation and caudal fin regeneration.” All changes made to the text are highlighted in yellow in the marked copy of revised manuscript (line 33).

4.

foxd3-positive blastemal cells

In the response letter, the author confirm that they “specified in the new version of our manuscript that *fox3*⁺ cells did not proliferate upon amputation” and later “*foxd3*⁺ cells are already present in intact fin fold and this number is not increased upon amputation”.

However, the abstract still contains a previous conclusion about a discovery of “a novel *foxd3*-positive blastemal cell population”. It is confusing given that the key feature of blastemal cells is their highly proliferative and less differentiated character, required for regeneration.

The claim that *foxd3*-positive cells can be considered as blastemal cells is not supported by evidence.

In general, I have an impression that several main conclusions/interpretations, especially in the abstract, should be revised according to the results.

We agree with the comment of the reviewer and accordingly modified the conclusions/interpretations. Therefore, throughout the manuscript, we replaced “...*foxd3*-positive blastemal cell population...” by “...*foxd3*-positive cell population within the regenerating fin...”. All changes made to the text are highlighted in yellow in the marked copy of revised manuscript (lines 32; 35; 347-348; 378).